



# A modular wind profile retrieval software for heterogeneous Doppler lidar measurements

Anselm Erdmann[1] and Philipp Gasch[1]

[1]Karlsruhe Institute of Technology, Institute of Meteorology and Climate Research Troposphere Research, Kaiserstraße 12, 76131 Karlsruhe, Germany

**Correspondence:** Anselm Erdmann (anselm.erdmann@kit.edu) and Philipp Gasch (philipp.gasch@kit.edu)

**Abstract.**

Retrieving wind profiles from Doppler lidar radial velocities requires processing software tools. The heterogeneity of Doppler lidar types and data acquisition settings, as well as scan patterns applied for wind profiling, make wind profile processing challenging. Addressing this challenge, a new modular open-source wind profile retrieval software is presented: the

Atmospheric Profile Processing toolKIT (AtmoProKIT). The software calculates quality controlled wind profiles from heterogeneous Doppler lidar data, i.e. independent of the system type, data acquisition settings or the scan pattern applied. Ingestion of heterogeneous data is enabled by the definition of a standardized level 1 data format for the measurements, from which level 2 wind profiles are retrieved. Processing flexibility is enabled through the combination of modular processing steps in module chains. Modifications are possible by individually arranging modules, adding calculation modules or adjusting processing pa-

rameters. The documentation of the processing steps in the result's metadata ensures the traceability of the results. A standard module chain is presented, which allows for straightforward wind profile retrieval for common Doppler lidar measurement scenarios without the need for coding. The results provided by the standard module chain are validated against radiosondes for three common Doppler lidar systems in differing atmospheric conditions. AtmoProKIT is provided as open-source Python code and includes demonstration examples, allowing for an easy use and future collaborative modification.

# 1  Introduction

Our understanding of atmospheric processes and the ability to forecast the weather rely on observations. Wind profile observations are crucial and present a missing link in the global observation system (Baker et al., 1995, 2014). Doppler lidars provide an established technique for range-resolved wind measurements using laser radiation (Werner, 2005). Coherent Doppler lidars measure the Doppler shift of the returned laser light, scattered by aerosols or other particles, compared to the outgoing laser

light. Thereby, Doppler lidars estimate the velocity of the scatterers in beam direction at different ranges along the beam. Assuming the scatterers are advected with the wind, the Doppler lidar measures the projection of the wind in beam direction, also called radial velocity. State-of-the-art Doppler lidars measure wind in a range O($100\,\mathrm{m}$)-O($10\,\mathrm{km}$) with high precision and accuracy.





Doppler lidars have become commercially available in the last decade and have seen increasing usage in research and in-
dustry, e.g. for atmospheric boundary layer research (Träumner et al., 2015; Adler et al., 2020) and wind energy applications
(Fernando et al., 2019). Existing Doppler lidars use various laser sources (pulsed, continuous), laser pulse characteristics (e.g.
wavelength, pulse energy, pulse length, pulse repetition frequency) and data processing settings (e.g. acquisition bandwidth,
FFT processing length, spectra accumulation time, number of range gates, range gate length, range gate spacing) (Werner,
2005). The ongoing development of lidar systems with different characteristics has resulted in instrument-specific heteroge-
neous data products and formats.

Additionally, Doppler lidars are used for a variety of application scenarios, including network wind profile retrievals (Wag-
ner et al., 2022), turbulence estimation (Bonin et al., 2017) and multi-Doppler setups for 2D flow retrieval (Träumner et al.,
2015) or virtual towers (Bell et al., 2020). The different scenarios require individual lidar and scan setups, which may addition-
ally depend on location and time. Thereby, an additional layer of heterogeneity is introduced, since the lidar data from each
application often requires unique algorithms for the specific evaluation tasks.

In this study, the heterogeneity created by different systems and applications, including laser, data acquisition and scan
pattern settings is addressed as *heterogeneous Doppler lidar data*.

One of the most common application scenarios for Doppler lidars is wind profiling, i.e. obtaining vertically resolved wind
vector information using a single Doppler lidar (e.g. Drew et al., 2013; Baker et al., 2014; Päschke et al., 2015, and references
therein). Wind profiles are easily interpretable and needed to study the atmospheric dynamics at the measurement site. Further,
wind profiles can be compared to and assimilated in numerical weather prediction (NWP) models (Newsom and Banta, 2004;
Kawabata et al., 2014; Pentikäinen et al., 2023).

To retrieve wind vectors (u, v, w components) from the radial velocities measured by the lidar, the lidar beam is scanned
through differing azimuth directions. At least three differing azimuth directions are required, and typically one or more eleva-
tion angles are used for scans focused on wind profile retrievals. The change in the measured radial velocity with azimuth is
then exploited to retrieve wind vectors using the established *velocity azimuth display* (VAD) (Browning and Wexler, 1968) or
*volume velocity processing* (VVP) methods (Waldteufel and Corbin, 1978; Boccippio, 1995). By performing the retrieval at
multiple distances along the beam, corresponding to multiple heights, remotely sensed vertical profiles of the wind vector (i.e.
wind profiles) up to multiple kilometres height become available (Baidar et al., 2023; Mense et al., 2024). In general, wind
retrievals become more challenging under weak signal conditions outside the planetary boundary layer (Baidar et al., 2023,
and references therein).

Retrieving wind profiles from heterogeneous Doppler lidar data is not straightforward, since system setup and scan pattern
vary depending on site characteristics and investigation aim. Besides the system heterogeneity discussed above, a multitude
of scan patterns exist for wind profiling. Trade-offs between the different scan patterns include the speed of execution, the
availability of the lidar signal or the vertical resolution. An influential parameter determined by the choice of the scan pattern is
the atmospheric volume probed by the lidar. The usage of multiple consecutive scans, corresponding to temporal aggregation,
is also possible in wind profile retrievals.



Both the probed volume and the aggregation time have a strong influence on retrieval accuracy due to the effect of atmospheric turbulence in the probed volume. As part of the retrieval, homogeneous flow is assumed within the volume probed by the lidar. Violations of this assumption result in retrieval error (Gasch et al., 2020), which becomes larger for shorter averaging times and steeper scans, i.e. closer to the vertical (Rahlves et al., 2022; Robey and Lundquist, 2022). Additional factors can also induce an error in the retrieval: Reasons include but are not limited to individual lidar effects and system errors (e.g. an insufficient noise compensation), range ambiguity due to high pulse repetition frequency or the influence of local conditions (e.g. topography or blocked sectors)

While wind profile retrieval from Doppler lidar data has been state-of-the-art in recent decades, it is often conducted using instrument-, setup- or even scan-specific software (Neto and Castelao, 2023). A common methodology and processing software is desirable to overcome complications when processing heterogeneous lidar data from multiple systems operated in variable setups. Additionally, comparability and traceability in wind profile processing is needed, especially if heterogeneous data from multiple systems is processed, e.g. for ingestion in data assimilation for numerical weather prediction models. Examples include processing of Doppler lidar data from network setups, currently operated in research environments (Kunz et al., 2022; Wagner et al., 2022), but likely also included in operational weather service networks in the near future (Kayser et al., 2021).

To our knowledge, an open-source software able to process a broad range of heterogeneous Doppler lidar data and provide quality controlled wind profile retrievals is missing up to date. Based on the need for reliable wind profile retrievals from heterogeneous Doppler lidar data, the present contribution presents the Atmospheric Profile Processing ToolKIT (AtmoProKIT), a new modular wind profile retrieval software. The software allows for wind profile retrievals independent of the type of Doppler lidar or scan pattern used. The processing architecture is designed in a modular way and conducts a chain of concatenated processing steps, which are arranged in calculation modules. This modular architecture allows for rapid user interaction with the processing chain without the need for coding. Thus, the user can configure the processing chain according to the individual needs and extend the chain with own algorithms. The Python code is provided to the scientific community as open-source software[1].

The software architecture is designed to

- handle heterogeneous data from different types of Doppler lidars with different data acquisition and scan settings, without major configuration modifications. Retrieving wind profiles from heterogeneous Doppler lidar data is enabled by a common data format definition, described in Sect. 2.

- enable wind profile retrievals in an easy-to-use and flexible way. To achieve this goal, the modular software architecture allows for easy user interaction and modification of the processing steps using so-called module chains presented in Sect. 3.

- provide quality-controlled and traceable wind profile retrievals, ready for further processing. A standard configuration applicable for a variety of instruments and conditions is presented in Sect. 4 and validated in Sect. 5.

---

[1]Preliminary access during the review: https://bwsyncandshare.kit.edu/s/8Yx4d9S9dmoJLjg





## 2 Enabling wind profile retrievals for heterogeneous Doppler lidar data

To retrieve the three-dimensional wind vector from Doppler lidar measurements, at least three radial velocity measurements from different, linearly independent, directions are necessary. Radial velocity measurements in different directions are achieved by scanning. Vertically resolved wind profiles are obtained through the range-resolved measurement capabilities of lidars, i.e.

the wind vector retrieval is performed at multiple altitudes using the respective radial velocity measurements.

In principle, a number of scan patterns are suitable for wind profile retrievals. Options include

- azimuthal scans at constant elevation, also called *plan-position-indicator* scans (PPIs), or sectors thereof;

- reduced but faster step-and-stares in orthogonal azimuth directions, also called *Doppler-beam-swinging* (DBS);

- varying elevation scans at constant azimuth directions, also called *range-height-indicator scans* (RHI);

- arrangements of fixed-direction stares, e.g. in step-and-stare approaches such as the six-beam method (Sathe and Mann, 2013);

or combinations thereof. Trade-offs regarding the suitability of the scan pattern for wind profiling exist. On the one hand, DBS scans provide faster repeat times and step-and-stare allow for longer signal accumulation, increasing the availability of the lidar signal. On the other hand, PPI scans provide better azimuthal resolution and RHI scans allow for measurements closer to the

ground, improving the vertical resolution.

The common base methodology for retrieving wind vectors is a least-squares fit of the radial velocity measurements. The required calculations are presented in detail in Appendix C. The calculations minimize the radial velocities' squared deviations from the estimated wind vector, based on an assumed homogeneous wind field in the volume probed by the lidar during scanning.

### 2.1 Minimizing the wind vector retrieval error requires maximized data availability

The wind vector retrieval error is determined by the applicability of the wind field assumed in the retrieval, by measurement errors, and by numerical errors during the calculation.

The wind vector retrieval is subject to the assumption of a homogeneous wind field inside the scan volume explored by the radial velocity measurements. Deviations from the homogeneity assumption, e.g. due to turbulence or other atmospheric

variability, introduce a retrieval error even without radial velocity measurement error (Bingöl et al., 2009; Gasch et al., 2020; Rahlves et al., 2022; Robey and Lundquist, 2022). Both scan setup and aggregation time have a strong influence on retrieval error. Steep measurements closer to the vertical result in a larger retrieval error (Teschke and Lehmann, 2017; Rahlves et al., 2022), making the inclusion of shallow scans desirable. If appropriate scan elevation angles are used, retrieval errors due to turbulence typically average out within a $10\,\mathrm{min}$ to $30\,\mathrm{min}$ span, depending on atmospheric conditions (Rahlves et al., 2022;

Robey and Lundquist, 2022). Therefore, a tradeoff between retrieval accuracy versus temporal resolution exists.

In addition to retrieval errors introduced by the assumption of homogeneous flow, erroneous radial velocity measurements by the lidar will also introduce retrieval errors. Hence, distinguishing reliable from unreliable radial velocity measurements is





a crucial challenge in the post-processing of Doppler lidar measurements. Filtering with the carrier-to-noise ratio (CNR) is a common method to improve measurement reliability (Päschke et al., 2015). Instead of the CNR some instruments provide a

signal-to-noise ratio (SNR), which can be used alternatively. In the following, only the term CNR is used. The CNR is not a completely reliable indicator. On the one hand, reliable measurements may be available even at low CNR. On the other hand measurements at high CNR may still be unreliable. Potential causes of erroneous measurements despite high CNR include second-trip echoes due to range ambiguity when using high pulse repetition frequencies (Päschke et al., 2015; Bonin et al., 2017), returns from flying objects, or hard target returns from terrain, as well as laser or data acquisition problems. As an un-

desired side effect of simple CNR threshold filters, many usable measurements are removed (Steinheuer et al., 2022). Hence, more advanced filters have been introduced over time (Bonin et al., 2017; Steinheuer et al., 2022; Päschke and Detring, 2024). Advanced filter approaches aim at considering only reliable radial velocity measurements on the one hand, while ensuring a broad data availability on the other hand. In some approaches, a priori information on wind field coherence based on climatology is used (Baidar et al., 2023). The trade-off between radial velocity measurement quality versus availability depends on the

individual use case, and may also be laser and scan pattern dependent. There is a need for an easy implementation of various radial velocity measurement filtering approaches for different lidar systems and operation scenarios.

In principle, the least-squares fit of the radial velocities for wind vector retrieval is applicable to arbitrary scans. However, additional retrieval errors arise if the beam directions of the radial velocity measurements are not sufficiently dispersed to ensure a reliable calculation of the resulting wind vector. In this case, the robustness and accuracy of the retrieved wind vector

is not ensured (Boccippio, 1995). The condition number (CN) is a measure for the robustness of the beam dispersion, whereby the condition number increases if the beam dispersion is closer to collinearity. Large condition numbers become an issue for reduced sector scans or scans close to the vertical axis (Wang et al., 2015; Päschke et al., 2015).

Overall, the issues of flow homogeneity assumption violation, radial velocity reliability, and condition number make it desirable to utilize the maximum number of measurements available for the retrieval. Maximizing the number of considered

measurements allows for the smallest assumption violation, due to the beneficial effect of averaging. Similarly, the largest possible data basis allows for better evaluation of measurement reliability and enables the largest possible beam dispersion.

In the following, the number of measurements used for the retrieval is maximized through the definition of retrieval volumes. Within each retrieval volume, measurements are considered independent of their scan pattern origin. To enable utilizing measurements from different scan patterns, a harmonized data format allows for ingestion of lidar data into the retrieval process

independent of their origin.

## 2.2 Maximizing the data availability through the definition of a retrieval volume

Typical scan patterns for the purpose of determining wind profiles include DBS, step-and-stare, PPI or RHI scans. If the scans employ equal heights of the range gate centres (typically the case for DBS and PPI scans), the calculation of wind vectors at specific height levels could in principle be aligned to the specific configuration. In Fig. 1, the beams on the cone represent an

eight-beam step-and-stare scan with five range gates. In this scan, each range gate centre is mapped to the same height above ground for all azimuth positions. In contrast, RHI scans include elevation angle changes and thereby do not sustain the height



above ground in different beams. As an example, an 18-beam RHI scan is visualized in Fig. 1. Retrieving wind vectors is not possible any more using range gate-based retrievals. Overcoming this limitation, the approach presented here takes scans into consideration independent of the scan geometry applied.

Maximizing the data availability requires the consideration of all available radial velocity measurements, independent of the scan type or range gate settings. The issue of changing absolute range gate heights is also present if the lidar position is not stable or not aligned in a horizontal plane, as for lidars operated on ships or aircraft (Zentek et al., 2018; Gasch et al., 2023). Changing absolute range gate heights for different laser beam positions can be overcome by binning the measurements with respect to the height above ground. In Fig. 1, the height bins are highlighted with the same background colour. In this way,

each measured radial velocity is associated with a height bin. In the following, the term *retrieval volume* describes one height bin during one temporal aggregation interval. Retrieval volumes can but need not be limited in their horizontal extent, i.e. the distance from the lidar. The definition of retrieval volumes enables the consideration of all measurements, independent of the scan type or range gate settings.

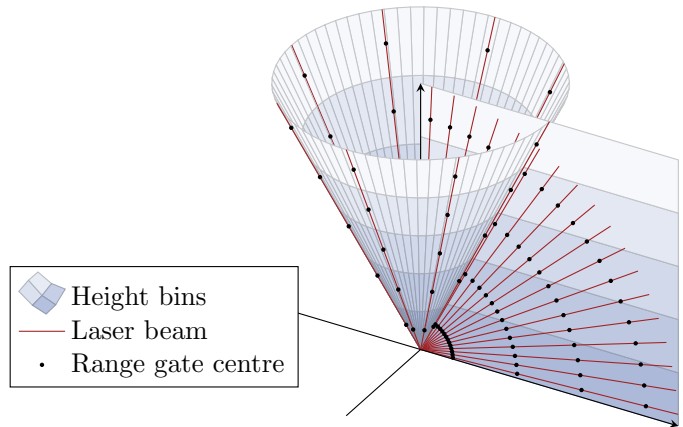

**Figure 1.** Visualization of typical scan patterns employed for wind profile retrieval. The respective range gate centres of all beams in the cone are mapped to the same heights, as typical for PPI and DBS scans. In contrast, the associated height of the range gate centres in RHI scans depend on the elevation angle, which is displayed on the plane. Retrieving the wind vector based on height bins enables the consideration of all measurements within the considered time period and, therefore, maximizes the data availability.

## 2.3   Level 1: A harmonized data format to enable maximized data availability

Doppler lidars provide measurements in a device-specific data format (level 0). The intended applicability of the radial velocity-based retrieval (Sect. 3) for various types of Doppler lidars and scan patterns requires a harmonized level 1 data format. A level 1 dataset supplies level 0 data for one requested instrument and a specified time span in a defined format, independent of the device-specific level 0 data format. Initially, level 0 data of the processed instrument is harmonized using a unified level 1 data structure, which serves as a common basis for the processing steps of the wind profile retrieval. The data is stored in dedicated

level 1 files, each comprising the measurements of one instrument for one day.





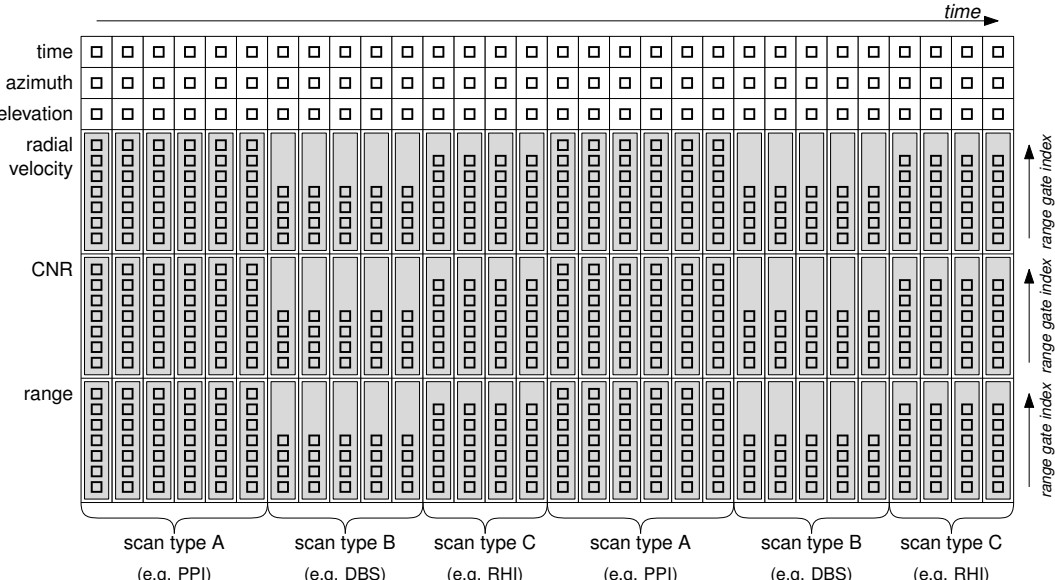

**Figure 2.** Level 1 format for one Doppler lidar. Using the dimensions *time* and *range gate index* enables the representation of different scan types and different settings. Scan type A could be a PPI scan with six beam positions and seven range gates, scan type B a DBS scan with four range gates, and scan type C an RHI scan with four different elevations and six range gates.

## Level 1 data structure

The level 1 data structure incorporates the typical netCDF format introduced by Rew and Davis (1990). It is widely used for observational data at level 0, although not all instruments provide netCDF files. In netCDF, variables are specified with dependence on dimensions. Multidimensional variables are an essential part of netCDF.

Typical level 0 Doppler lidar measurements rely on the dimensions time and range. The time coordinates contain the time-stamps of the measurements, the range coordinates the distance of the range gate centres from the instrument. The time coordinate can differ between instruments at level 0 and will then also differ at level 1.

For each time and range, radial velocity, CNR, and possibly more measurements are provided alongside azimuth and elevation positions. Building a harmonized level 1 dataset from such level 0 datasets could be achieved by concatenating level 0

datasets along the time dimension if the range dimension remains the same. However, possibly different range gate settings for differently configured subsequent scans (see Sect. 2.2) prevent simple concatenations of all measurements. Hence, a more general format is required.

The wind retrieval software uses, therefore, the dimensions *time* and *range gate index* at level 1. Using the range gate index as a dimension is independent of the real ranges, which are instead stored as a variable depending on time and range gate index.

The redundancy of storing range as a function of scan type and range gate index for every time step is necessary to enable the concatenation of scans with different range gate lengths or a different number of range gates. The scan type with the highest





number of range gates determines the overall size of the range gate index dimension. Unused ranges are filled with NaN in case a scan type uses less than the maximum number of range gates. Mandatory variables (associated dimensions in brackets) are *time* (*time*), *azimuth* (*time*), *elevation* (*time*), *radial velocity* (*time*, *range gate index*), *CNR* (*time*, *range gate index*), and *range*
(*time*, *range gate index*). The SNR can be used instead of the CNR.

The level 1 format is not limited to the introduced variables. Further variables, such as the Doppler spectrum width (depending on time and range gate index), aerosol backscatter (depending on *time* and *range gate index*), inclination angles (depending on *time*), or others, can be added.

**Exemplary representation of different scan types**

Figure 2 visualizes the level 1 data format. The associated Doppler lidar conducts the scan types A, B, and C in a loop. In the *time* dimension, the format follows the lidar data acquisition timestamp. In the *range gate index* dimension, scan type A has the largest number of range gates and determines the size.

Scan type A comprises six laser beam positions and seven range gates. For each position, the corresponding timestamp, the azimuth angle, and the elevation angle are stored. The two-dimensional variables *radial velocity*, *CNR*, and *range* have the
shape of an array for each timestamp. Scan type B comprises only five beam positions and four range gates. Additionally, the *range* variable may change for different times in scan type B, since the range gate length and spacing of the vertical beam are typically different for DBS. Scan type C comprises six range gates and four beam positions. While the range variable is typically constant within an RHI scan, the absolute height of each measurement has to be calculated using the variables *elevation* and *range* (see Sect. 2.2).

**3 Wind profile retrieval (level 1 to level 2)**

This section describes the wind profile retrieval processing. Wind vectors are calculated based on the harmonized level 1 radial velocity measurements (Sect. 2.3), which serve as input data. Level 2 represents wind vectors per time and height aggregation bin (i.e. the wind profiles when looking at all heights), as detailed in Sect. 2.2. A new modular software architecture enables a flexible adaptation of the processing steps based on the individual conditions and user needs.

**3.1 Level 2 data format**

Level 2 wind vectors are represented by the wind vector direction components *u*, *v*, and *w*. One wind vector is calculated per bin of time and height, also termed retrieval volume. Therefore, the level 2 dimensions are *time* and *height*. Additionally, the *nv* dimension is used to specify the lower and upper bin bounds for both coordinates according to the CF metadata conventions (Eaton et al., 2023). Figure 3 shows a visualization of the level 2 data format. The dimensions are *time*, *height*, and *nv*. Besides
the coordinate variables *time* and *height*, the boundary variables *time_bnds* and *height_bnds* are mandatory. The temporal and height resolution is specified by the user in the settings (Sect. 3.3).

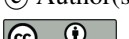



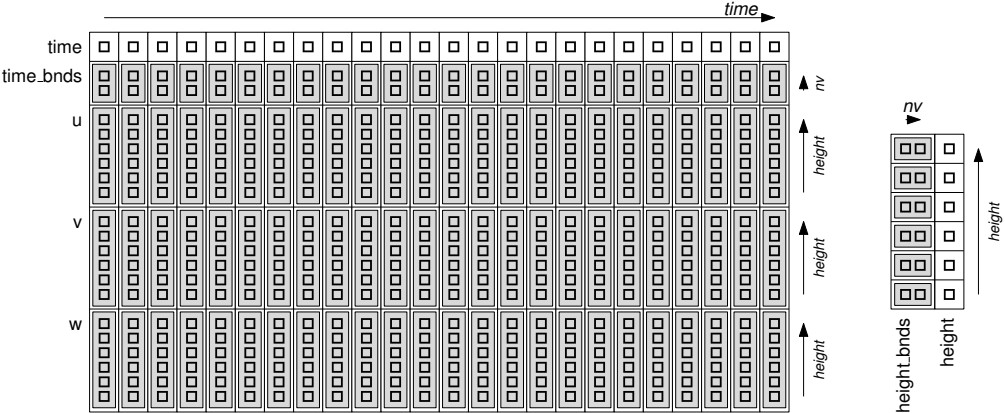

**Figure 3.** Level 2 format for one Doppler lidar. The coordinates are *time* and *height*. The variables *u*, *v*, and *w* describe the resulting wind vectors in the associated direction components.

## 3.2 Modular software architecture for flexible wind profile retrievals

A software architecture capable for processing measurements from various systems benefits from flexibility in the processing algorithm, to adapt for individual user needs or special application cases (Sect. 1).

Hence, the software is split into modules, which each perform specific processing tasks. Each module performs a small, self-contained part of the overall processing. The desired processing of level 1 radial velocities to level 2 wind vectors is conducted by the combination of modules, which are specified in a module chain. The modular architecture enables fast re-configuration of the processing chain, as well as extensions by other modules. New developments will be added to the code repository in the future.

**Module chain description**

The level 1 dataset (Sect. 2.3) and an empty level 2 dataset form the initial input for the module chain. Each calculation module receives both, the level 1 and the level 2 dataset, as inputs. The calculations performed inside a module consider a set of specified variables from one or both levels. After a module's calculations are completed, the results are added as variables to the corresponding level 1/level 2 dataset or replace existing variables. Both datasets are returned, replace the originally received

datasets, and serve as input for the next module in the chain. In case a dataset has not changed, the respective input dataset remains unchanged.

Figure 4 shows a schematic module chain arrangement. Calculation modules include algorithms for retrieving wind vectors, indicating non-reliable measurements (e.g. based on CNR or CN values), applying filters (e.g. level 1 data flagging of azimuth or elevation ranges), but can also include algorithms for the indication of specific conditions (e.g. flagging of precipitation).

In addition to calculation modules, also export modules can be integrated in a module chain. Export modules export variables,



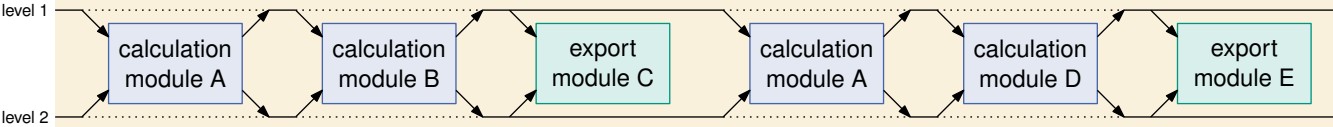

**Figure 4.** Schematic module chain configuration. The level 1 and the level 2 dataset can be changed by the modules according to the specified variables flow. The modules are executed sequentially and consider the calculated results of the respective predecessor module. Solid lines indicate datasets' information flow. Datasets to be replaced are displayed dotted.

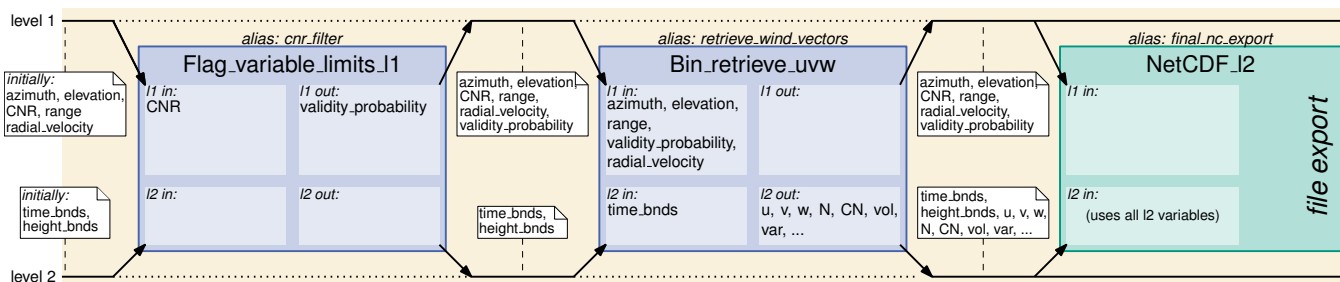

**Figure 5.** Exemplary module chain for a simple retrieval with CNR threshold filtering and netCDF export. For each module, considered input variables and written output variables are given by the inset boxes (*l1 in*, *l1 out*, *l2 in*, *l2 out*). The annotations outside the module boxes indicate the data contained in the level 1 and level 2 dataset. Coordinate variables (*time* (l1, l2), *range gate index* (l1), *height* (l2)) are not indicated.

e.g. as files or visualizations, but do not change the dataset itself. Required input and available output variables are specified for each module. The arrangement of the modules can follow the individual needs. The modules can be arranged in an arbitrary order, as long as the necessary input variables are available in the dataset. Modules can also be used multiple times in the same module chain.

An exemplary module chain for a wind profile retrieval with a simple CNR filter is provided in Fig. 5, the corresponding module chain file is shown in Appendix A1. The level 1 dataset initially contains the variables received from the instrument, which are *CNR*, *range*, and *radial_velocity*. The level 2 dataset is initially empty and contains only the level 2 bin bounds according to the selected bin specification. Coordinates (*time, range gate index* for level 1; *time, height* for level 2) describing the dataset are also contained but not visualized. Both datasets are fed into the *Flag_variable_limits_l1* module, which uses the

*CNR* variable and adds the variable *validity_probability* to the level 1 dataset. The *validity_probability* variable contains an acceptance (1.0) or rejection (0.0) value for each bin according to a user-defined threshold. The level 1 input variable *CNR* and the level 1 output variable *validity_probability* are renamings of the default variable names for the use of the *Flag_variable_limits* module as a CNR filter (for details, see Appendix A1).





The second module (*Bin_retrieve_uvw*) calculates the level 2 wind vectors based on the level 1 variables *validity_probability*,
*radial_velocity*, *range*, *azimuth*, and *elevation*. The resulting wind vectors for each bin (*u*, *v*, *w*) and ancillary variables are added
to the level 2 dataset. The level 1 dataset is returned without changes. Finally, the *NetCDF_l2* export module writes the level 2
dataset into a netCDF file.

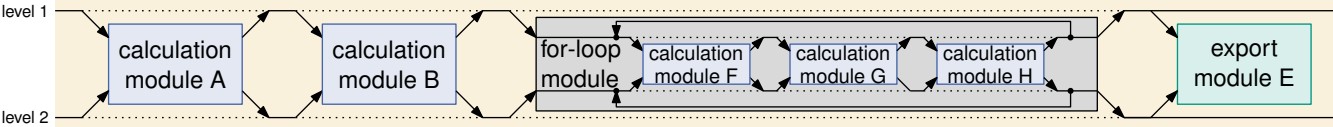

**Figure 6.** Module chain with a loop module. The calculation modules F, G, and H are integrated in the for-loop module which causes the
repeated execution of the hierarchically integrated modules.

**Loops in module chains**

Further modules are loop modules, which enable a repetition of modules. Modules that should be repeated in a loop are
hierarchically integrated in the loop modules. In Fig. 6, a for-loop module is executed after the calculation modules A and B.
It contains the modules F, G, and H, which are executed multiple times in a for-loop. The number of iterations is specified in
the module chain file. Additionally, a while-loop module is available, which repeats the loop as long as a condition regarding
a specified level 2 variable is met. It is possible to create hierarchical configurations with loop modules within a loop module.

**Module parametrization and variables renaming**

The introduced modular retrieval configuration enables the user to arrange the calculation modules according to the individual
needs. Sophisticated retrievals could require an individual parametrization of the modules, depending on their position in the
module chain. Therefore, an alias name has to be given to each module in a module chain. This ensures the possibility of
individual module parametrization, even if the same module is contained multiple times in a module chain (see module A in
Fig. 4).
Furthermore, a module arrangement can require to change input/output variable names of modules to enable e.g. bifurcations
in the processing. Renaming the default input and output variables of a module prevents the replacement of a variable by this
module in case it is already contained in the dataset. Therefore, renaming of the variables can be specified in the module chain.
This renaming and the arrangement of the modules in the module chain determine the variable flow.

**3.3 Retrieval configuration**

The software is executed with Python. However, for an easy usability, the configuration of the module chain and settings occurs
in two plain text files. The module chain file (*.mc) contains the arrangement of the modules. The settings file (*.ini) contains
the level 2 vertical and temporal resolution, directory paths, and processing parameters. A settings file for a configuration of





the simple module chain (Fig. 5) is given in Appendix A2.

The software architecture is designed for application cases with three types of user: (1) users without programming experience
with need for straightforward Doppler lidar wind profile retrievals, (2) users with need for detailed configurations for special
circumstances or instrument conditions, and (3) methodology developers. Hence, the software architecture provides several
optional configuration and extension possibilities.

(1) Users with the need for straightforward Doppler lidar wind profile retrievals can resort to the standard module chain
and settings for common application cases described in Sect. 4. The performance of this standard configuration is validated
in Sect. 5. The user has to provide the level 1 dataset and select the instrument as well as the time span to be processed. The
required information can be provided in an interactive dialogue, or as command line arguments (for bash scripts), alternatively.
Common Doppler lidar types (WTX, WLS200s, StreamLine) are already included in the settings file, others can be added
through supplying CNR acceptance thresholds for the respective device type. Except for minor adaptations such as setting
import and export directories, no changes are required to the provided settings file. Instructions are given in the readme file in
the code repository and an example dataset which can be processed is provided as part of this publication[2].

(2) User with the need for specific configurations can change the configuration in the module chain file and in the settings
file. The JSON-formatted module chain representation allows for adding or omitting modules (see Appendix A1). Renaming
the input and output variables of a module enables the user to specify the desired variable flow. By supplying module alias
names, it is possible to execute the same module multiple times with individual parameter settings on each execution (see
module A in Fig. 4). Detailed descriptions of the available modules and configuration instructions for users with modification
purposes are available in the readme file and in the manual.

(3) Developers can insert their own modules. Abstract base classes are available for calculation and export modules and
ensure the interoperability with the other modules. A developer has to implement the mandatory abstract methods, such as
providing information on the required and generated variables. The algorithm has to be inserted in a calculation module, which
receives the current level 1 and level 2 datasets as arguments. After the conduction of the algorithm, the revised level 1 and
level 2 datasets have to be returned. Instructions for developers can be found in the manual.

## 3.4    Program execution

After the user has completed the configuration of module chain and settings, the program can be started by executing `python
AtmoProKIT.py`.
The selection of module chain, settings, instrument, and the time span which should be processed can be supplied by the
user interactively or using a bash script. Level 1 input data is requested by the program based on the desired level 2 time span.
As modules could require an extended time span to take also previous and subsequent time into consideration, the resulting
required level 1 time span is calculated by requesting the required input time span for a given result time span for each module
in inverse order. The corresponding method is implemented for each module and returns the required input time span for
the requested time span, which is required for the successive module. Loop iterations have to be considered in the time span

---

[2]Preliminary access during the review: https://bwsyncandshare.kit.edu/s/8Yx4d9S9dmoJLjg



calculation. To avoid infinite required time spans, a maximum number of iterations has to be specified for while-loops, which is considered for the determination of the required time span. The finally resulting time span is requested from the level 1 data source. The received level 1 time span together with an empty level 2 dataset serve as initial input for the execution of the module chain. Finally, all modules are executed according to the module chain in the given order.

### 3.5 Tool interoperability and processing traceability

The retrieval software needs level 1 datasets as input (Sect. 2.3) and provides level 2 datasets, containing the wind profiles, as output. To ensure a broad interoperability with other tools, netCDF (Rew and Davis, 1990) is supported for level 1 import and level 2 export. For an easy handling within Python, xarray, introduced by Hoyer and Hamman (2017), is used as data type to represent the level 1 and the level 2 datasets internally. Xarray uses a data representation similar to netCDF and enables simple conversion from and into netCDF and NumPy arrays. Using NumPy arrays enables significant speedup through vectorized operations (van der Walt et al., 2011). Level 1 data can be provided as netCDF files, each comprising the measurements of one instrument for one day. The provided level 1 files need to contain all variables required for the further processing. Through the netCDF format, also pre-processing of level 1 files with other software tools is possible. The direct integration of other formats than netCDF for level 1 import requires the implementation of an import module, which creates the required xarray dataset.

Besides tool interoperability, which is ensured through using netCDF, traceability of the processing is important. The CF conventions require an audit trail for data modifications (Eaton et al., 2023). Due to the flexibility of a module chain, adding the tool name to the metadata would not ensure a sufficient traceability. Therefore, the level 2 dataset also contains the conduction history of the data modifying calculation modules, including the parameter values. Each module execution is added, also within loops. After the netCDF export, the module chain execution history is available as an attribute in the exported netCDF file. This documentation ensures the reproducibility of the conducted processing steps.

### 3.6 Results of a simple module chain retrieval with CNR filtering

A simple wind profile retrieval using a fixed CNR threshold filter (module Flag_variable_limits_l1) and a common wind vector retrieval method, considering the robustness of the retrieved vectors and iteratively removing outliers (module Bin_retrieve_uvw), can be implemented in the new architecture using the module chain shown in Fig. 5. This module chain comprises the established mechanisms for wind vector retrievals and serves as a basis for further refinement in Sect. 4. Even though the Bin_retrieve_uvw module applies an iterative mechanism to remove unreliable measurements during wind vector estimation, the initial CNR filter has the main influence on the data availability and quality.

Figure 7 shows the resulting wind profiles of the simple module chain exemplary for one day with a bin resolution of $10\,\mathrm{min}$ temporally and $100\,\mathrm{m}$ vertically. The associated CNR levels are shown in Fig. 8. On the one hand, the CNR of $-25\,\mathrm{dB}$ applied in this example serves as a reliable threshold for the WLS200s system used, i.e. there are no obvious retrieval outliers present. Outliers due to individual erroneous radial velocity measurements are also prevented by the iterative removal of measurements with more than $3\,\mathrm{m\,s^{-1}}$ residual in the least-squares wind vector fit of the radial velocities used in the retrieval process. On the



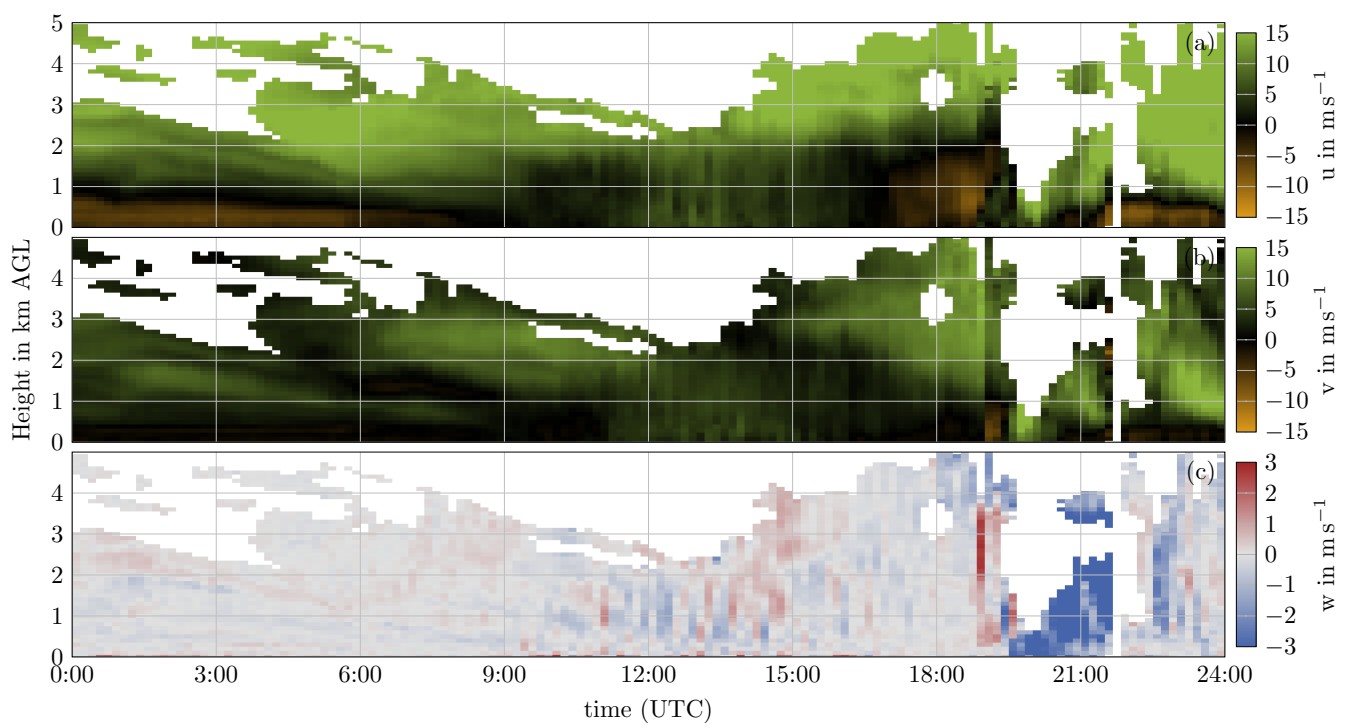

**Figure 7.** Wind profiles retrieved from a WLS200s Doppler lidar considering only high-confidence radial velocity measurements with a CNR of at least $-25\,\mathrm{dB}$. The DBS and RHI scans were conducted at Fischerbach (Germany) on $11^{\mathrm{th}}$ July 2023.

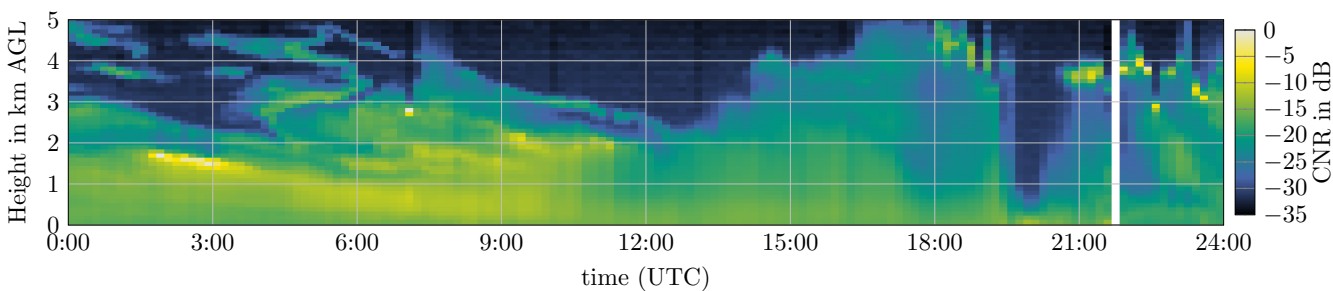

**Figure 8.** Median of the CNR values of all measurements available per bin, i.e. without the application of CNR filtering. Heavy precipitation caused a weak signal at 20:00 UTC. The gap at 20:40 UTC is caused by an instrument failure.



other hand, the data availability provided by this simple wind profile retrieval is limited. The CNR values indicate a low but possibly usable signal in areas where no wind vector retrieval is available, due to the strict CNR filter applied.

Selecting a lower CNR threshold of $-30\,\mathrm{dB}$ exploits these regions, but some of the additionally retrieved wind vectors are erroneous despite the iterative outlier removal, as indicated by outliers in the retrieved wind vectors (Fig. E1). The share of erroneous wind vectors might be acceptable for visual evaluations. However, trustworthy wind vectors are essential in case of automated processing in subsequent evaluations or data assimilation. Hence, an extended module chain for common conditions with more sophisticated processing is introduced in Sect. 4 and validated in Sect. 5.

## 350   4   Extended standard module chain for common conditions

This section presents an extended module chain suitable for common application cases. The so-called *standard module chain* allows wind profile retrieval without deep methodological knowledge. The associated module chain file and settings file, suitable for various Doppler lidar systems and typical application cases, are provided in the code repository. The standard module chain is designed to provide high availability of retrieved level 2 wind vectors while maintaining high retrieval quality. To 355  achieve this, the modules are arranged in the module chain configuration listed in Table 1. The corresponding variable flow is given in Table B1. This standard module chain configuration is suitable for mixtures of different scans to maximize the number of used measurements (see Sect. 2.2).

### 4.1   Functional description of the standard module chain

At the beginning, the standard module chain conducts a first guess of the level 2 wind profile retrieval based on high-quality level 1 data, which is controlled through an initial (instrument type-dependent) strict CNR filter threshold. Despite the strict CNR threshold, not all measurements, and thus retrievals, may be reliable (see Sect. 2). Therefore, the first guess level 2 retrieval is further controlled using a 2D median filter, before interpolating into a so-called *confidence background*. Based on the filtered and interpolated level 2 confidence background, expected level 1 radial velocity measurements are calculated.

Trustworthy level 1 data with lower CNR values are then included if the radial velocity is within a maximum acceptable radial velocity deviation tolerance. Repeating the procedure in an iterative procedure refines and extends the available level 2 data. Finally, post-processing and data export is conducted.

    The following paragraphs provide a description of the processing procedures and reasoning implemented in the standard module chain. The processing parameters and thresholds contained in the modules are described. Users can adjust the thresh-
olds and processing parameters in the settings file, if needed.

**Principal level 1 filtering: Consideration of measurements**

During the initial level 1 elevation filtering (module (1)), measurements are excluded to avoid sampling biases or less robust beam constellations (see Sect. 2.1). For example, for low retrieval height bins and shallow scan elevations, numerous radial ve-





**Table 1.** Standard module chain for common conditions. Modules (7.1) to (7.5) are calculated in three iterations within the for-loop (7).

| No | Module | Module type | Description |
|---|---|---|---|
| (1) | Flag_elevation_limits_l1 | calculation | l1 elevation angle filter |
| (2) | Flag_horizontal_distance_limits_l1 | calculation | l1 maximum horizontal distance from lidar filter |
| (3) | Multiply_variables | calculation | l1 consideration (results from modules (1) and (2)) |
| (4) | Flag_variable_limits_l1 | calculation | strong l1 CNR filter |
| (5) | Flag_variable_limits_l1 | calculation | weak l1 CNR filter |
| (6) | Bin_retrieve_uvw | calculation | uvw retrieval on time and height bins |
| (7) | For_loop | for-loop | 3 iterations |
| (7.1) | Median_filter_l2 | calculation | l2 median filter |
| (7.2) | Extrapolation_uvw_l2 | calculation | l2 NaN filling extrapolation |
| (7.3) | Flag_Vr_using_uvw_l2_to_l1 | calculation | l1 filter accepting data within the l1 background tolerance |
| (7.4) | Multiply_variables | calculation | l1 filter combination (results from modules (7.3) and (5)) |
| (7.5) | Bin_retrieve_uvw | calculation | uvw retrieval on time and height bins |
| (8) | Bin_statistics_l1_to_l2 | calculation | provide CNR and Doppler spectrum width information to l2 |
| (9) | Set_quality_flags_qu_qv_qw_l2 | calculation | set quality flags if the retrieved wind vector is not NaN |
| (10) | Get_ff_dd_from_uv_l2 | calculation | calculate horizontal wind speed and direction |
| (11) | Plot_universal | export | plot quicklooks |
| (12) | Remove_variables | calculation | remove ancilarry l2 variables |
| (13) | NetCDF_l2 | export | save l2 netCDF file |

locity measurements are mapped to the same height bin (see Fig. 1). Thereby, an imbalance between a high number of shallow
scan elevations (surface-near measurements) and a low number of steeper scan elevations can be created. Such an imbalance
negatively impacts the vertical wind vector calculation and is, therefore, undesirable. Hence, very shallow scan elevations (de-
fault: below 15°) are excluded from the calculations by module (1). Module (2) limits the horizontal (not along-beam) distance
for which radial velocity measurements are considered (the default value is 3 km), to reduce the impact of inhomogeneities in
the far surroundings. The measurements to be considered in principle are labelled by module (3), which combines the con-
sideration flags (0 or 1) of modules (1) and (2) by multiplication. In the further processing, level 1 measurements are only
considered if the consideration flag is 1.

**First guess wind profile retrieval and initial quality assessment**

To obtain a reliable first guess of the wind profiles, a strict CNR filter (the default threshold for Leosphere WLS200s is $-25\,\mathrm{dB}$)
is applied in module (4). It adds a validity flag for each level 1 measurement to the level 1 dataset. Similarly, a weak CNR
validity flag (default threshold $-30\,\mathrm{dB}$ for Leosphere WLS200s) is provided by module (5) and stored for later use. Wind
vectors are then retrieved in module (6). The least squares fit of the wind vector is calculated for each bin (see Appendix C)
by considering those level 1 radial velocity measurements where both the consideration flag and the validity probability are 1.





If level 1 measurements deviate too much from the fit, they are removed, and the wind vector calculation is repeated without them. The default accepted deviation tolerance is $\pm\,3\,\mathrm{m\,s^{-1}}$. The retrieval procedure is repeated iteratively until all remaining

measurements are within the accepted tolerance, or an insufficient number remains and the wind vector is rejected.

To achieve resilient wind vectors, the retrieval quality is assessed, and the retrieved wind vector is rejected, if one of the quality indicator conditions is violated. The quality indicators are the condition number (default threshold: 8); the volume enclosed in the convex hull spanned by the unit vectors in the laser beam directions (default threshold: 0.042, corresponding to about $2\,\%$ of the unit hemisphere explored); the absolute number of measurements (default threshold: 12); and the share

of measurements that contributed to the least squares fit (default threshold: 0.2), in relation to all considered measurements (see module (3)). The condition number is an indicator for the robustness of the laser beam dispersion (see Sect. 2.1). For unbalanced scan patterns (e.g. an overwhelming majority of vertical stares alternating with few PPI/DBS measurements), the condition number may be high, yet wind profile retrieval may be possible due to the PPI/DBS measurements. The enclosed volume threshold allows for inclusion of such unbalanced scans despite a high condition number. The absolute number of

measurements prevents retrievals based on very few measurements, where a few erroneous measurements can have strong influence. The relative number of measurements prevents retrievals in conditions where random noise is the dominant signal.

Up to this point, the module chain is the simple chain (discussed in Sect. 3.6), extended by the principal level 1 filtering. The further steps improve the data basis considered for wind vector retrieval to also enable retrievals in regions with low CNR values.

**Lower CNR acceptance threshold based on confidence background**

CNR values provide a first-order indication of data quality. However, even at low CNR reliable measurements may be available in some circumstances, depending on atmospheric and lidar system conditions (Sect. 2.1).

To extend the availability of retrieved wind vectors at level 2, measurements with lower CNR should also be considered if they are trustworthy. Therefore, a level 2 (u, v, w) wind profile confidence background is calculated based on the level 2 wind

profile first guess. Since a high reliability of the confidence background is crucial, the first guess is filtered with a median filter (default: 3 time bins, 5 height bins, i.e. depending on the selected resolution) in module (7.1), to remove potentially remaining outliers. Subsequently, the confidence background is extrapolated in module (7.2). In Fig. 9, the extrapolated confidence background is displayed for the same setting as in Fig. 7.

Based on the level 2 wind profile confidence background, the expected level 1 radial velocity is calculated for every mea-

surement. Level 1 radial velocities which are within a tolerance (default: $\pm\,3\,\mathrm{m\,s^{-1}}$) of the radial velocity expected from the confidence background are considered (7.3), even at lower CNR, as long as the CNR is above the weak threshold indicated by module (5). Thereby, the level 1 data basis is extended with quality controlled radial velocity measurements exhibiting a lower CNR.

Module (7.5) performs the wind vector retrieval again, but takes all measurements accepted by modules (7.3) and (5) into

account. The increase in the level 1 data availability resulting from more accepted measurements results in a larger availability of retrieved wind vectors at level 2.



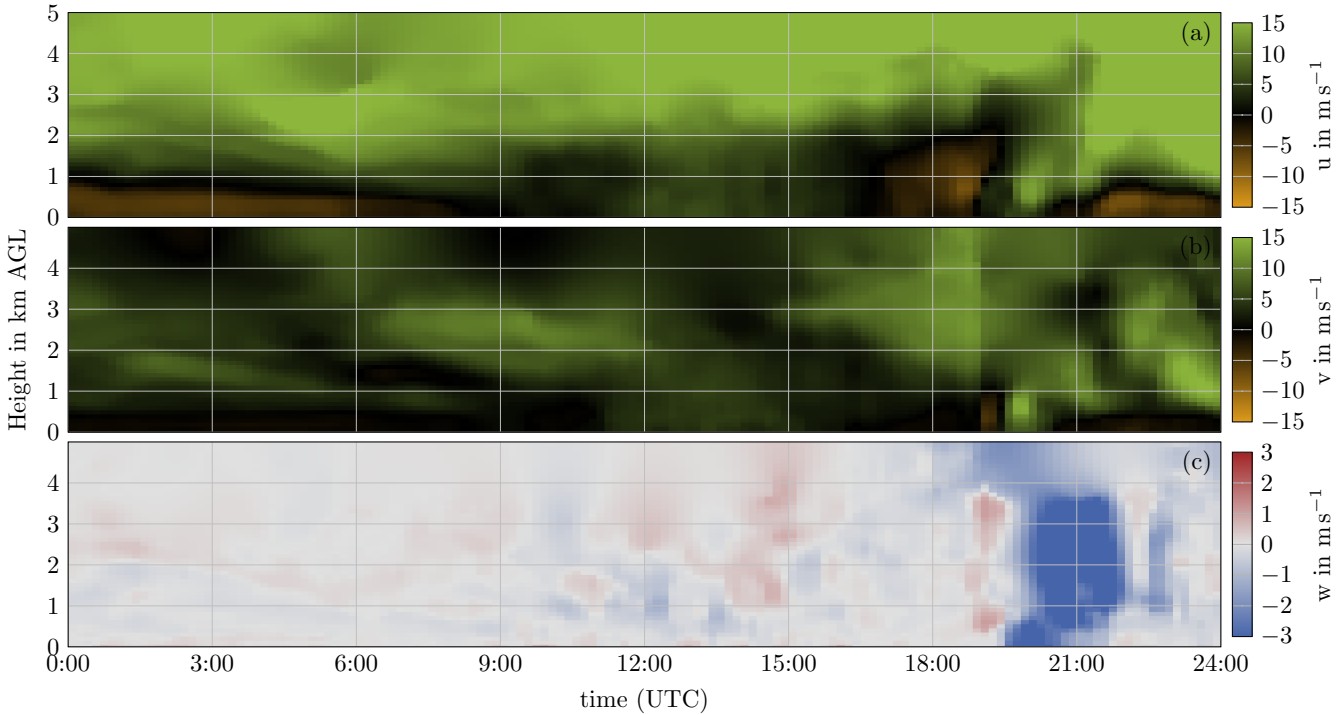

**Figure 9.** Wind profile confidence background. The confidence background is calculated based on the first guess wind profiles using an additional median filter and subsequent extrapolation. Radial velocities that deviate within a specified tolerance from the expected radial velocity according to the confidence background are considered in the next iteration, if a weaker CNR threshold is met.

**Confidence background quality control**

A good estimation of the confidence background is crucial, since it determines the acceptable radial velocity range during subsequent iterations. While the iterations allow for improvement and refinement of the confidence background, the quality of
the used radial velocities and retrieved wind vector must be ensured in every iteration. As such, accepting noisy radial velocity measurements, which match the tolerance of the confidence background by coincidence, needs to be avoided. Therefore, the wind vector retrieval (module (7.5)) uses a minimum threshold on the share of accepted measurements in relation to all available measurements within each bin. The default threshold used in the standard configuration is 0.2, which reliably avoids accidental fitting of noise (Sect. 5).

**Iterative refinement**

Modules (7.1) to (7.5) are repeated two more times as specified in the for-loop (7). During the second and the third iteration, the wind profiles retrieved during the respective previous iteration are used instead of the first guess. The confidence background is refined and improved over the iterations. The magnitude of the benefit of the single iterations varies, depending on the instru-



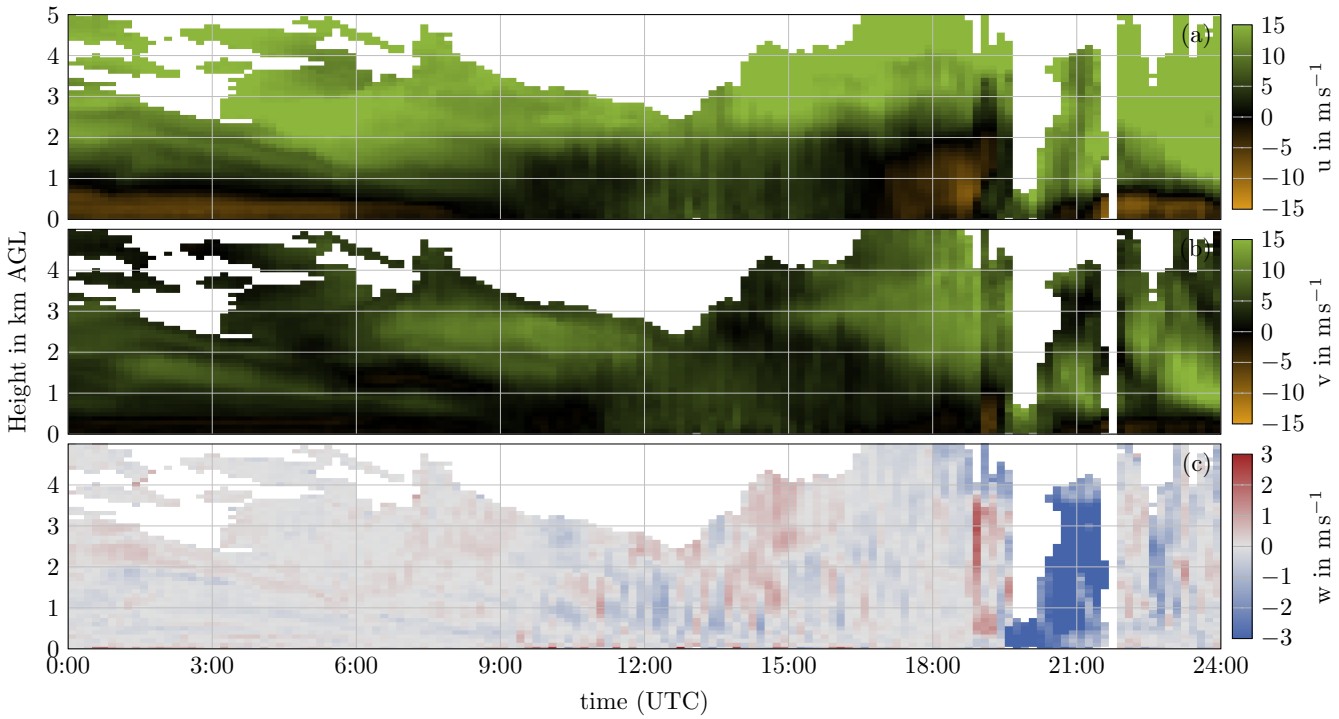

**Figure 10.** Final wind profiles. All measurements with a CNR of at least $-30\,\mathrm{dB}$ are also considered if the radial velocities deviate below $\pm\,3\,\mathrm{m\,s^{-1}}$ from the radial velocity expected from the confidence background. The data availability is improved compared to the first guess, particularly in gaps.

ment type and current atmospheric conditions. An evaluation of the iteration benefit for ten Doppler lidars with quantification of the additionally retrieved wind vectors follows in Sect. 4.2.

## 4.2 Demonstration of the standard module chain for common conditions

The advantages of the standard module chain are showcased in comparison to the simple chain presented in Sect. (3.6). Additionally, the benefit of the iterative approach is demonstrated using three-months of measurements gathered using ten Doppler lidar systems during the Swabian MOSES 2023 campaign. A validation of the standard module chain retrieval quality with radiosondes follows in Sect. 5.

**Advantages compared to the simple chain**

The final result of the retrieval is shown in Fig. 10. The number of retrieved wind vectors increases from 4929 (first guess, Fig. 7) to 5553, and the additional values are plausible. To illustrate the effect of the confidence background in retaining plausible measurements using the weaker CNR filter, Fig. E1 shows the result of modules (1) to (6) conducted with a CNR threshold of $-30\,\mathrm{dB}$ (i.e. directly, without confidence background plausibility check).



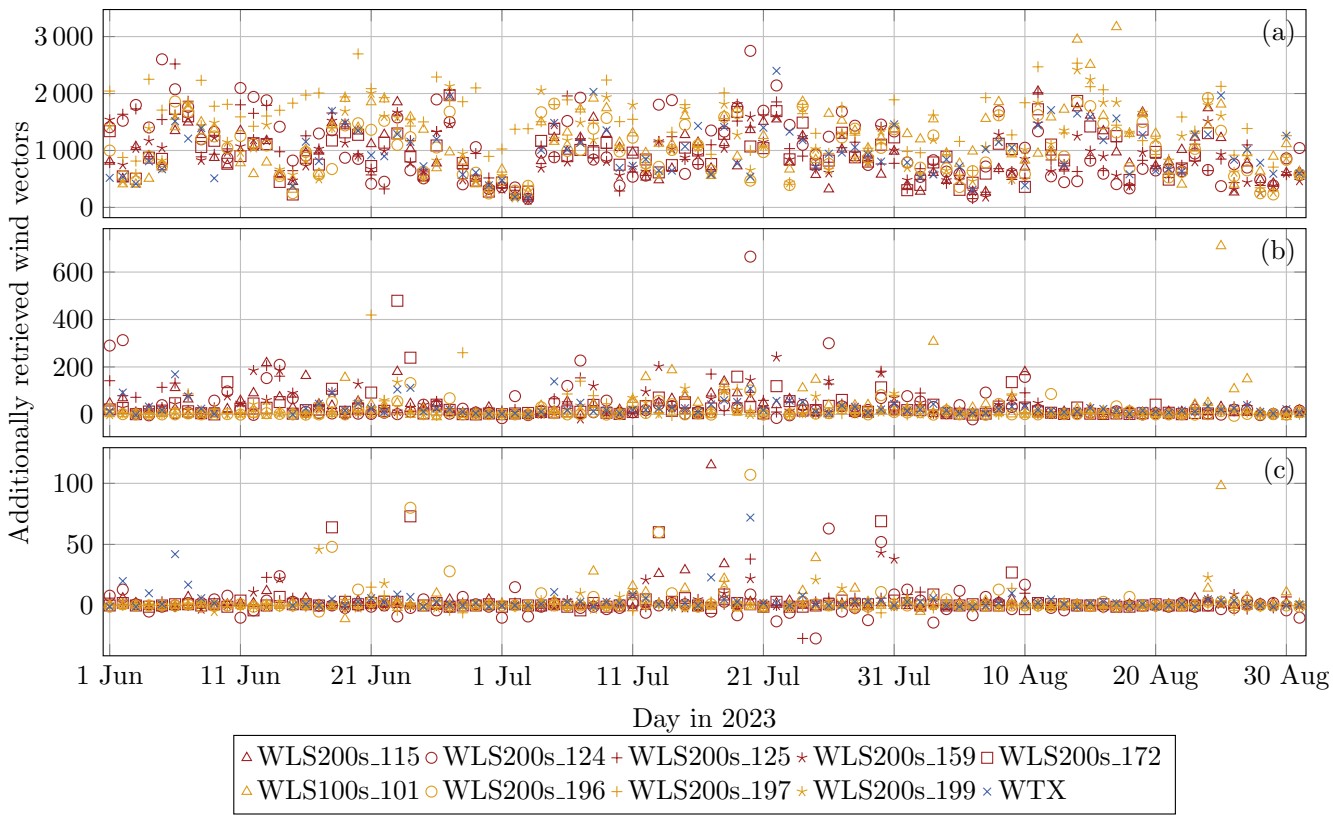

**Figure 11.** Number of additional wind vectors gained through the three iterations of modules (7.1) to (7.5). Results are provided on a daily basis for ten Doppler lidars operated in the southern Black Forest (Germany) and neighbouring regions in Switzerland during a three-month period in summer 2023. The difference is displayed between (a) the first guess and iteration 1, (b) iteration 1 and iteration 2, and (c) iteration 2 and iteration 3.

In sum, a wind vector is retrieved for 5430 bins in the simple $-30\,\mathrm{dB}$ CNR threshold retrieval, compared to 5553 for the standard module chain retrieval. At the upper bound of the available data, slightly more wind vectors are retrieved with the standard module chain. More importantly, however, the edge areas are not contaminated with outlier wind vectors when using the standard chain retrieval. The prevailing reduction of unreliable wind vectors using the standard chain retrieval is essential for automated processing of wind data, both for statistical analysis and for input in models. Hence, the introduced standard module chain presents an advantage.

**Iterative wind vector availability enhancement**

Maximized level 1 data availability (Sect. 2.2) forms the basis for an increased level 2 wind vector availability. The iteratively refined confidence background extends the level 1 data availability and, therefore, also the availability of retrieved level 2 wind vectors. Figure 11 shows the difference in the number of wind vectors retrieved per day between different numbers of iterations





for ten Doppler lidars operated during a three-month period as part of the Swabian MOSES 2023 experiment. The WLS200s
with the numbers 115, 124, 125, 159, and 172 conducted RHI scans between 0° and 60° elevation in four directions (azimuth
0°, 90°, 180°, 270°) every five minutes and DBS scans at 60° elevation in the remaining time. The other Doppler lidars were
operated with various settings and scans of type DBS, RHI, PPI, and fixed-direction stares. The first iteration (Fig. 11 (a))
retrieves considerably more wind vectors for all instruments compared to the first guess. An additional number of 1440 wind
vectors available per day corresponds to an average availability increase of $1000\,\mathrm{m}$, given the $10\,\mathrm{min}$ temporal and $100\,\mathrm{m}$
vertical resolution. Between subsequent iterations (Fig. 11 (b) and (c)), a significant increase is observed for a few days and
stations. The number of additionally retrieved wind vectors decreases compared to the respective preceding iteration. Whether
a higher availability from more iterations justifies the increased calculation effort can be decided by the user in the individual
application case.

## 5   Validation of the standard module chain

The wind profiles obtained using the standard module chain and settings, operated with $10\,\mathrm{min}$ temporal resolution and $100\,\mathrm{m}$
vertical resolution, are validated against radiosonde measurements for three different Doppler lidar systems at different loca-
tions. The Doppler lidar systems included for evaluation cover a Leosphere WLS200s, a Lockheed Martin WindTracer (WTX)
and a Halo Photonics StreamLine XR+. The radiosondes used as a validation reference were released in the near surroundings
of the respective Doppler lidar. The geographic and temporal distribution of the station data covers a wide range of atmospheric
conditions, ensuring representativeness of the validation.

### 5.1   Data basis used for validation

The WLS200s wind profile retrieval results are compared with 180 ascending radiosondes between June and August 2023 at
the MeteoSwiss site Payerne, Switzerland (46.81° N, 6.94° E). Heterogeneous scans of type DBS, step-and-stare, and sector
PPIs are used for wind profile retrieval (Fig. E2 (a)). The radiosondes were launched regularly at 11:00 UTC and 23:00 UTC,
thereby covering both daytime and nocturnal conditions.

The WTX wind profile retrievals are evaluated against 131 ascending radiosondes launched during the Swabian MOSES
2023 experiment at Villingen-Schwenningen, Germany (48.06° N, 8.49° E). The lidar conducted PPI scans (elevations of
3°, 30°, 60°) alternating with RHI scans in directions from 0° elevation to 90° elevation (Fig. E2 (b)). Radiosondes were
launched three-hourly during eight intensive observation periods targeting the initiation of thunderstorms, thereby covering
pre-convective and convective conditions with assumed strong spatial flow variability.

For the validation of the StreamLine XR+, publicly available lidar measurements (Schmithüsen et al., 2024) from the Neu-
mayer Station, Antarctica (70.67° S, 8.27° W) are processed. The retrieved wind profiles are compared to 212 radiosondes
usually released at 11:00 UTC between 4[th] January 2024 and 26[th] November 2024 (Schmithüsen, 2022). The lidar conducted a
12-beam step-and-stare pattern (also termed VAD pattern for Halo systems, Fig. E2 (c)), alternating with vertical stares. In the



following, these VAD scans are used. Due to the remote Antarctic location, strong and vertically sheared katabatic flows and challenging lidar measurement conditions with low aerosol concentrations are covered.

Before quantitative evaluation, initial visual analysis of the wind profiles obtained using the WLS200s at Payerne and the WTX at Villingen-Schwenningen shows that physically plausible results are provided by the standard module chain. Hence, for these systems, there is no need for a reconfiguration of the standard module chain or the adaptation of thresholds. The results obtained from the Neumayer lidar show a peculiar artifact: At weak SNR, very frequent vertical winds with $-1\,\mathrm{m\,s^{-1}}$ are observed in the absence of horizontal wind (i.e. horizontal wind speeds of approximately $0\,\mathrm{m\,s^{-1}}$). Likely, a non-uniform radial velocity noise spectrum for measurements in weak SNR conditions causes the observed effect. The observed phenomena and the applied solution are discussed in Appendix D. The ingestion of the suspicious radial velocities is prevented through an additional level 1 data filter applied for Neumayer Station, which excludes radial velocities between $-2\,\mathrm{m\,s^{-1}}$ to $-0.35\,\mathrm{m\,s^{-1}}$ for SNR values below $-22\,\mathrm{dB}$.

## 5.2 Retrieval validation for wind speed

A number of reasons complicate validation of the lidar wind profile retrievals with radiosonde measurements, since a point-based in situ vertical profile is compared to a volume-based remote sensing retrieval. Non-homogeneous wind in time or space causes differences, as the retrieval represents an average over the aggregated time ($10\,\mathrm{min}$) and height while the radiosonde crosses a $100\,\mathrm{m}$ height bin in approximately $20\,\mathrm{s}$. Radiosondes may yield non-representative measurements and are advected with the flow, hence increasing spatial distances between the wind profiles may be present, especially at higher altitudes. Lidar retrieved wind profiles suffer from retrieval errors due to flow inhomogeneity, depending on the scan patterns and atmospheric conditions (Gasch et al., 2020; Rahlves et al., 2022; Robey and Lundquist, 2022). Besides the differences in measurement characteristics, both systems may also suffer from direct measurement errors due to system imperfections.

Fig. 12 shows the comparison of the radiosonde measurements with the corresponding retrieved horizontal wind speed for all three stations. WLS200s (Payerne) and WTX (Villingen-Schwenningen) exhibit good agreement over the full wind speed range. The StreamLine XR+ (Neumayer) exhibits lower data availability but slightly higher agreement with the radiosondes. Deviations remain fairly constant in magnitude over the full wind speed range for all stations.

The WLS200s comparison at Payerne offers a representative data basis under various atmospheric conditions due to the regular operational launches without targeting of special atmospheric phenomena. Good agreement with a small positive bias of the wind speed retrieval is observed.

The deviations between radiosonde and lidar measurements are slightly larger at Villingen-Schwenningen and more measurements under high wind speed conditions are included. Potentially, the increased scatter is related to the targeting of the radiosondes towards convective and pre-convective conditions, i.e. more spatial variability and less representative measurements. For the WTX, one radiosonde profile is omitted from the comparison (11[th] July 2023, 21:32 UTC) due to non-representative measurement conditions. An analysis of the corresponding meteorological situation reveals that the radiosonde was launched into a passing meso-scale precipitating system and thus encountered strongly non-representative conditions during its ascent and drift away from the lidar.





**Figure 12.** Agreement of the retrieved horizontal wind speeds with radiosonde measurements. An orthogonal distance regression is calculated for the Doppler lidars and radiosondes at Payerne (a, d, g), Villingen-Schwenningen (b, e, h), and Neumayer Station (c, f, i). The horizontal distance of the radiosonde to the lidar position (a-c), the retrieved vertical wind speed (d-f), and the occurrence in bins of $0.5\,\mathrm{m\,s^{-1}}$ (g-i) are colour-coded. For Villingen-Schwenningen, one radiosonde profile is omitted from the comparison due to non-representative measurement conditions (see text).



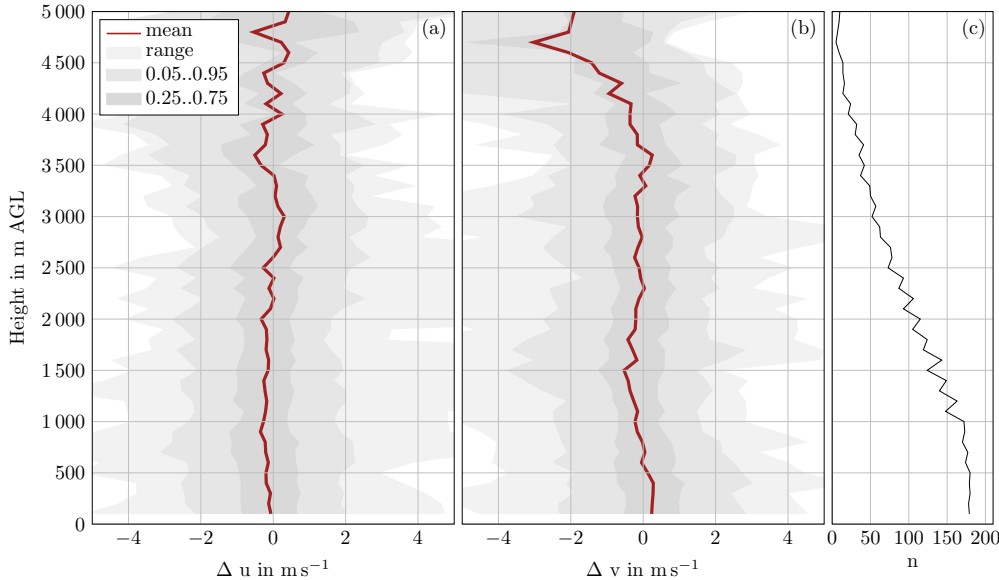

**Figure 13.** Horizontal wind component differences between radiosondes and WLS200s retrieval (10 min temporal resolution) at Payerne. The radiosonde wind components are subtracted from the retrieved Doppler lidar wind components. The shaded areas mark quantiles. The number of retrieved wind vectors n decreases with height.

Similarly, the generally more stable atmospheric conditions at Neumayer introduce less spatial variability and hence provide more representative measurements, in addition to reduced lidar retrieval error due to more homogeneous flow.

For all stations, the largest deviations are frequently associated with non-zero vertical winds retrieved by the lidar (Fig. 12 d-f). Especially comparisons with a negative vertical wind measured by the lidar show an increased scatter. The increased scatter
can be attributed to less homogeneous flow conditions, leading to less representative radiosonde measurements and increased lidar retrieval errors (Gasch et al., 2020; Rahlves et al., 2022; Robey and Lundquist, 2022). The retrieved negative vertical velocities are often associated with precipitation (snow, rain), which presents additional challenges for lidar measurements.

### 5.3   Retrieval validation for wind component profiles

For further validation including the vertical retrieval characteristics, the difference between the wind profiles retrieved from
Doppler lidar measurements and radiosonde measurements is analysed. The retrieved wind profiles are compared to the radiosondes by subtracting the radiosonde's u and v component from the respective component of the retrieved Doppler lidar wind profiles. The resulting differences are shown for Payerne (WLS200s) in Fig. 13, for Villingen-Schwenningen (WTX) in Fig. 14, and for Neumayer Station (StreamLine XR+) in Fig. 15.

For Payerne and Villingen-Schwenningen, the comparison shows good agreement between the wind profiles at altitudes
with sufficient data availability below 4 km, especially when considering the varying footprints and distances between remote sensing and in situ measurements. The interquartile distance is stable with altitude, highlighting the consistent retrieval quality





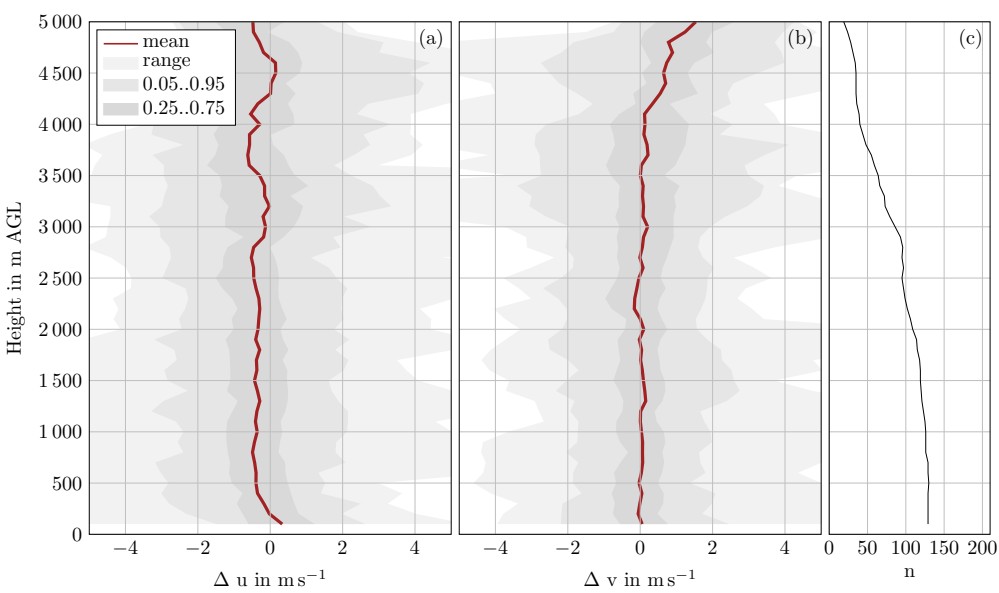

**Figure 14.** Horizontal wind component differences between radiosondes and wind profiles retrieved from the WTX Doppler lidar at Villingen-Schwenningen.

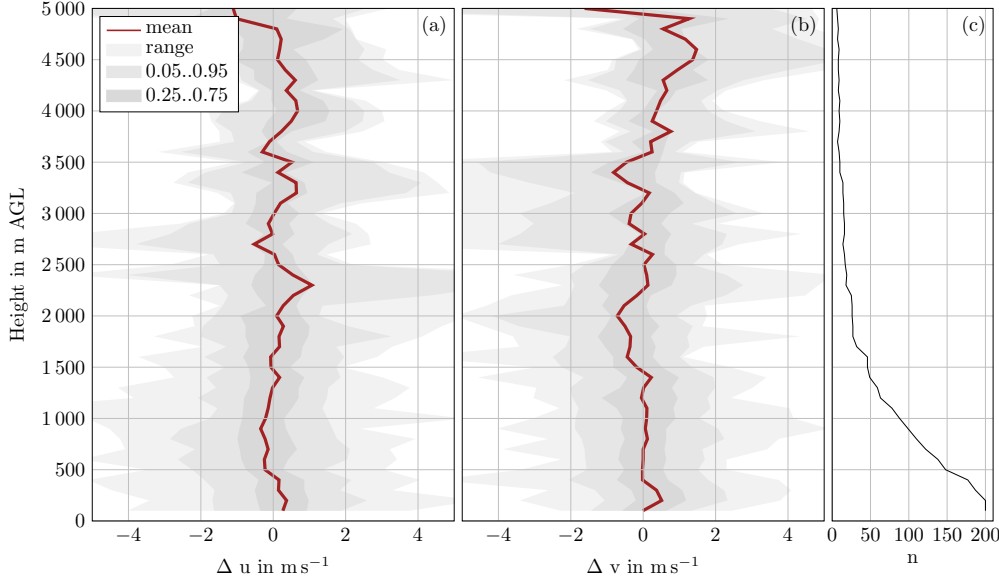

**Figure 15.** Horizontal wind component differences between radiosondes and the wind profiles retrieved from the StreamLine XR+ located at the Neumayer Station at 10 min temporal resolution. The number of available samples enables a reliable validation only at low heights.





above the boundary layer, where only a reduced backscatter from aerosols is available. Larger differences are observed above $4\,\mathrm{km}$, attributable to the insufficient sample size and large horizontal distances between the radiosondes and the Doppler lidar. The slight variation in the number of retrieved wind vectors with altitude for Payerne results from a different number of radial velocities mapped to the single height bins. Thereby, a higher wind vector availability for bins with more radial velocities is observed.

The results of the StreamLine XR+ lidar at Neumayer Station displayed in Fig. 15 also indicate a good agreement at the low heights, where the number of retrieved wind vectors is sufficient. Similar to the improved wind speed comparison at Neumayer, the interquartile range is also slightly reduced for the components at Neumayer. However, more vertical variability is observed due to the smaller sample size, related to the often low aerosol concentrations in Antarctica.

**The impact of temporal resolution**

A small systematic difference (bias) in the u component of about $0.4\,\mathrm{m\,s^{-1}}$ is evident for altitudes below $3\,\mathrm{km}$ at Villingen-Schwenningen, which is also detectable in the quartiles. No similar bias is observed in the v component, and also not in any component at Payerne or Neumayer Station. Hence, the difference is likely attributable to local flow or representativeness effects. The u-component shows a systematically higher mean values, possibly creating sampling artefacts due to the occurrence of gusts. Wind gusts cause an increased wind speed for a limited time. The retrieved wind vectors consider, however, the complete interval of temporal aggregation, which is $10\,\mathrm{min}$. The introduced consensus-based retrieval could smooth such gusts if the wind speed is significantly higher than during the major time of the temporal bin integration interval.

For comparison, the retrieval with a temporal resolution of $5\,\mathrm{min}$ is shown in Fig. E3. Using this higher resolution, the offset decreases slightly to about $0.3\,\mathrm{m\,s^{-1}}$ and shows less variation up to $4\,\mathrm{km}$ height. Interestingly, the increase in temporal resolution does not increase retrieval errors noticeably. Instead, the interquartile range is similar or even slightly reduced. Since the WTX scan pattern was completed within $5\,\mathrm{min}$, full scans are available even at this shorter averaging duration. Hence, the retrieval volume is explored equally well also for this temporal resolution. Nevertheless, even for full scans, one may expect less favourable averaging of retrieval error due to turbulence in the retrieval volume (Robey and Lundquist, 2022; Rahlves et al., 2022). The fact that no increase in retrieval error is observed for shorter retrieval times illustrates the suitability of the scan pattern and the retrieval algorithm. Further, the expected slight increase in retrieval error due to turbulence may be compensated by a smaller representativeness error in the comparison with the radiosonde. One reason may be the increased ability to capture short-term gusts for shorter retrieval periods (Steinheuer et al., 2022).

## 6 Conclusions

A novel modular software for the retrieval of wind profiles from heterogeneous Doppler lidar measurements is introduced.

To enable processing of measurements from various Doppler lidar systems, a standardized level 1 data format is used. The level 1 format provides data homogenization and enables subsequent retrieval of wind profiles using consistent processing routines, independent of the system origin. To maximize data availability, wind vectors are retrieved after binning the measured





radial velocities with respect to time and height. Thus, all available measurements from various data acquisition settings and
scan types can be exploited for the retrieval, independent of their origin.

The software architecture is designed in a modular way, enabling flexible adaptation without a need for coding. The data filtering and retrieval process is defined in module chains, in which the arrangement of modules specifies the processing steps and variables' flow. Each module performs a dedicated calculation with defined input and output variables, as well as required parameters. A configuration file is used to provide or modify parameter values required by the modules during operation.
The modular architecture enables a high flexibility for adaptions with reduced development effort. The software framework is provided as open-source Python code.

Thus, the software is suitable for three types of user:

1. Users with need for straightforward Doppler lidar wind profile retrievals can use the standard module chain configuration, which provides validated and quality controlled wind profiles for common conditions and multiple Doppler lidar systems.

2. Users with special investigation purposes or uncommon measurement parameters can configure the module chain and parameters according to their interests and needs.

3. Developers can contribute their own algorithms in new modules.

The delivered standard module chain for common conditions comprises an initial first guess of the wind profile with reliable measurements, based on a strict CNR filtering threshold. An iterative procedure adds measurements with a lower CNR, if
the measurements are within the expected range of the extrapolated previous guess, and if a sufficient share of measurements support the retrieved wind vector. Through the iterative procedure, quality controlled wind profiles are provided also for weak CNR conditions.

A validation of the retrieved wind profiles is conducted for three different Doppler lidar systems at different locations. Atmospheric conditions ranging from summer-time convective boundary layers to wintertime stable boundary layers in the
Antarctic are investigated. The comparison with radiosonde measurements reveals a high quality of the retrieved wind profiles for all investigated Doppler lidars and atmospheric conditions.

Overall, the modular software package provides capabilities to implement different retrieval scenarios for a wide range of users, Doppler lidar systems and applications. The introduced modular software architecture enables the flexibility to cover various application cases on the one hand, but ensures also a high-level traceability, which is essential for the processing history
of measurements used in models, on the other hand.

*Code availability.* The current version of the software can be accessed during the review under https://bwsyncandshare.kit.edu/s/8Yx4d9S9dmoJLjg. The software will be made permanently publicly available in the Helmholtz Codebase once accepted for publication. A snapshot of the version used for the validation will be made available once accepted for publication.





*Data availability.* Measurement data of the Swabian MOSES 2023 campaign will be published under the frame publication Handwerker
et al. (2025). The Doppler lidar measurements from Neumayer Station (Schmithüsen et al., 2024) and Payerne (Hervo and Coen, 2024) are
publicly available in the ACTRIS Cloudnet data portal https://cloudnet.fmi.fi.

## Appendix A: Software configuration files example

### A1 Module chain example

Module chains are configured JSON-formatted. Adding or omitting modules is possible through removing or inserting the
respective text blocks in the module chain file. An exemplary JSON-formatted representation for the module chain displayed
in Figure 5 is provided here.

```
[
    {
        "type":"calculation",
        "alias_name":"cnr_filter",
        "module_name":"Flag_variable_limits_l1",
        "rename_parameters": {
            "min_value":"cnr_threshold_dB"
        },
        "rename_level1_inputs": {
            "variable":"CNR"
        },
        "rename_level1_outputs": {
            "condition_met":"validity_probability"
        }
    },
    {
        "alias_name":"retrieve_wind_vectors",
        "type":"calculation",
        "module_name":"Bin_retrieve_uvw"
    },
    {
        "alias_name":"final_nc_export",
        "type":"export",
        "module_name":"NetCDF_l2"
    }
```





]

The module chain consists of three JSON objects, each representing one module (see Fig. 5).

The first module (alias name *cnr_filter*) implements a CNR filter by setting the flag variable *validity_probability* to 1.0 if the
CNR value is at least the threshold specified in the parameter *cnr_threshold*. Otherwise, the value of *validity_probability* is 0.0.
The calculation module used for the implementation of the CNR filter is *Flag_variable_limits*. This module uses *min_value*
as default parameter for the minimum threshold, a variable with the name *variable* as default level 1 input, and a variable
with the name *condition_met* as default level 1 output. Therefore, the default names for the parameter, the level 1 input,
and the level 1 output are renamed for the use as CNR filter. This renaming is possible by specifying *rename_parameters*,
*rename_level1_inputs*, and *rename_level1_outputs* for this module.

The second module (alias name *Bin_retrieve_uvw*) calculates the level 2 wind vectors using the level 1 radial velocities where
the value of *validity_probability* is 1.0. No renaming of variables or parameters is needed, as the default names are used.

The third module (alias name *NetCDF_l2*) exports the final level 2 dataset to a netCDF file.

Parameters are specified in the settings file (Sect. A2). The user is requested to insert not specified parameters in an interactive
dialogue before the module chain is executed.

## A2  Settings file example

```
[import]
level1_netcdf.level1_import_directory=/home/user/Level1
level1_netcdf.level1_variables=["radial_velocity","CNR","elevation","azimuth","range"]
[export]
global.level2_export_directory=/home/user/Level2
[parameters]
global.nrel_thresh_processing=0.2
global.cn_thresh=8
global.volume_thresh=0.042
global.variance_thresh=3
global.offset_first_bin_meter=-50
global.temporal_aggregation_seconds=600
global.vertical_aggregation_meter=100
global.max_height_meter=5050
global.max_radial_velocity_deviation_m_per_s=3
global.Nmin=12
global.validity_probability_acceptance_threshold=0.5
[parameters_instrument_type.WLS200s]
```



`global.cnr_threshold_dB=-25`





## Appendix B: Variable flow in the standard module chain





**Table B1.** Variable flow in the standard module chain. Coordinates and related variables are omitted in this listing. Read access (r) and write access (w) are indicated for the modules introduced in Table 1

| Level | Variable(s) | (1) | (2) | (3) | (4) | (5) | (6) | (7.1) | (7.2) | (7.3) | (7.4) | (7.5) | (8) | (9) | (10) | (11) | (12) | (13) |
|---|---|---|---|---|---|---|---|---|---|---|---|---|---|---|---|---|---|---|
| 1 | elevation | r | r | | | | r | | | r | | r | | | | | | |
| 1 | azimuth | | r | | | | r | | | r | | r | | | | | | |
| 1 | radial_velocity | | | | | | r | | | r | | r | | | | | | |
| 1 | CNR | | | | r | r | | | | | | | r | | | | | |
| 1 | range | | r | | | | r | | | r | | r | | | | | | |
| 1 | doppler_spectrum_width | | | | | | | | | | | | r | | | | | |
| 1 | consideration_elevation | w | | r | | | | | | | | | | | | | | |
| 1 | consideration_horizontal_distance | | w | r | | | | | | | | | | | | | | |
| 1 | consideration | | | rw | | | r | | | | | r | | | | | | |
| 1 | validity_probability | | | | w | | r | r/w | r/w | w | r/w | r | | | | | | |
| 1 | validity_probability_cnr_weak | | | | | w | r | | | | r | | | | | | | |
| 2 | u, v | | | | | | w | r/w | r/w | r | | w | | r | r | r | | r |
| 2 | w | | | | | | w | r/w | r/w | r | | w | | r | r | r | | r |
| 2 | N, CN, vol, var,... | | | | | | w | | | | | w | | | | | w | |
| 2 | cnr_all, Nall, ... | | | | | | | | | | | | w | | | | | |
| 2 | qu, qv, qw, qwind | | | | | | | | | | | | | w | | | | r |
| 2 | dd,ff | | | | | | | | | | | | | | w | | | r |





**Appendix C: Wind profile retrieval theory**

Calculating the least-squares fit to retrieve wind vectors from measured radial velocities is an established approach (e.g.
Päschke et al., 2015; Teschke and Lehmann, 2017; Zentek et al., 2018; Bell et al., 2020; Steinheuer et al., 2022; Gebauer and Bell, 2024). For the calculation of one wind vector $\boldsymbol{v}_\mathrm{p} = (u, v, w)^\mathrm{T}$, a projection to the multiple radial velocity measurements $\boldsymbol{V}_\mathrm{r} = (V_{\mathrm{r}1}, ..., V_{\mathrm{r}n})^\mathrm{T}$ is required, which is displayed in Equation (C1). The system of equations (C1) projects the wind vector components $u$, $v$, and $w$ to the radial velocities $V_{\mathrm{r}1}, \dots, V_{\mathrm{r}n}$ depending on the given angular position of the scanner (azimuth $\alpha$, elevation angle above the horizontal plane $\epsilon$).

$$\begin{pmatrix} \sin(\alpha_1)\cos(\epsilon_1) & \cos(\alpha_1)\cos(\epsilon_1) & \sin(\epsilon_1) \\ \vdots & \vdots & \vdots \\ \sin(\alpha_n)\cos(\epsilon_n) & \cos(\alpha_n)\cos(\epsilon_n) & \sin(\epsilon_n) \end{pmatrix} \cdot \begin{pmatrix} u \\ v \\ w \end{pmatrix} = \begin{pmatrix} V_{\mathrm{r}1} \\ \vdots \\ V_{\mathrm{r}n} \end{pmatrix}. \tag{C1}$$

This relation can be expressed in the following way:

$$\mathbb{G} \cdot \boldsymbol{v}_\mathrm{p} = \boldsymbol{V}_\mathrm{r}. \tag{C2}$$

This equation system is overdetermined for more than three measurements. Thereby, a retrieval of the wind vector is possible through minimizing the least-squares sum of the residuals.

The residuals $\boldsymbol{\delta}$ describe the deviation of the measured radial velocities from the wind vector $\boldsymbol{v}_\mathrm{p}$. The residuals are

$$\boldsymbol{\delta} = \mathbb{G} \cdot \boldsymbol{v}_\mathrm{p} - \boldsymbol{V}_\mathrm{r}. \tag{C3}$$

To receive the least squares fit of $\boldsymbol{v}_\mathrm{p}$, the sum of the squared residuals s has to be minimized. It is

$$s = \boldsymbol{\delta}^\mathrm{T} \cdot \boldsymbol{\delta}. \tag{C4}$$

This least-squares fit problem can be solved by multiplying the Moore-Penrose pseudoinverse (Päschke et al., 2015; Menke, 2012):

$$\mathbb{G}^+ = (\mathbb{G}^\mathrm{T}\mathbb{G})^{-1}\mathbb{G}^\mathrm{T}, \tag{C5}$$

from the left to Equation (C2), which finally results in

$$\boldsymbol{v}_\mathrm{p} = \mathbb{G}^+ \cdot \boldsymbol{V}_\mathrm{r}. \tag{C6}$$

An alternative calculation of the pseudoinverse using the matrices obtained with the singular value decomposition of $\mathbb{G}$ is less susceptible to errors for poorly conditioned $\mathbb{G}$ (Boccippio, 1995). Therefore, algorithms usually calculate the pseudoinverse (Menke, 2012):

$$\mathbb{G}^+ = \mathbb{W}_\mathrm{p}\mathbb{S}_\mathrm{p}^{-1}\mathbb{U}_\mathrm{p}^\mathrm{T}, \tag{C7}$$



where $\mathbb{W}_p$, $\mathbb{S}_p$, and $\mathbb{U}_p$ are submatrices of $\mathbb{W}$, $\mathbb{S}$, and $\mathbb{U}$, and

$$\mathbb{G} = \mathbb{U}\mathbb{S}\mathbb{W}^T \tag{C8}$$

is the singular value decomposition of $\mathbb{G}$. $\mathbb{S}_p$ is a diagonal matrix of the singular values of $\mathbb{G}$, which is $\mathrm{diag}(\lambda_1, \lambda_2, \lambda_3)$ for the present problem.

   $\mathbb{S}_p$ is also needed for the calculation of the condition number. Usually, $\mathbb{S}_p$ is calculated with $\lambda_1, \lambda_2, \lambda_3$ in descending order (Menke, 2012), which is also the case in the NumPy implementation[3] used in the present software. The condition number CN is the maximum singular value divided by the minimum singular value (Boccippio, 1995), which is

$$\mathrm{CN} = \frac{\lambda_1}{\lambda_3}. \tag{C9}$$

---

[3]https://numpy.org/doc/stable/reference/generated/numpy.linalg.svd.html, accessed 5th November 2024



## Appendix D: Extension of the standard module chain for Neumayer Station

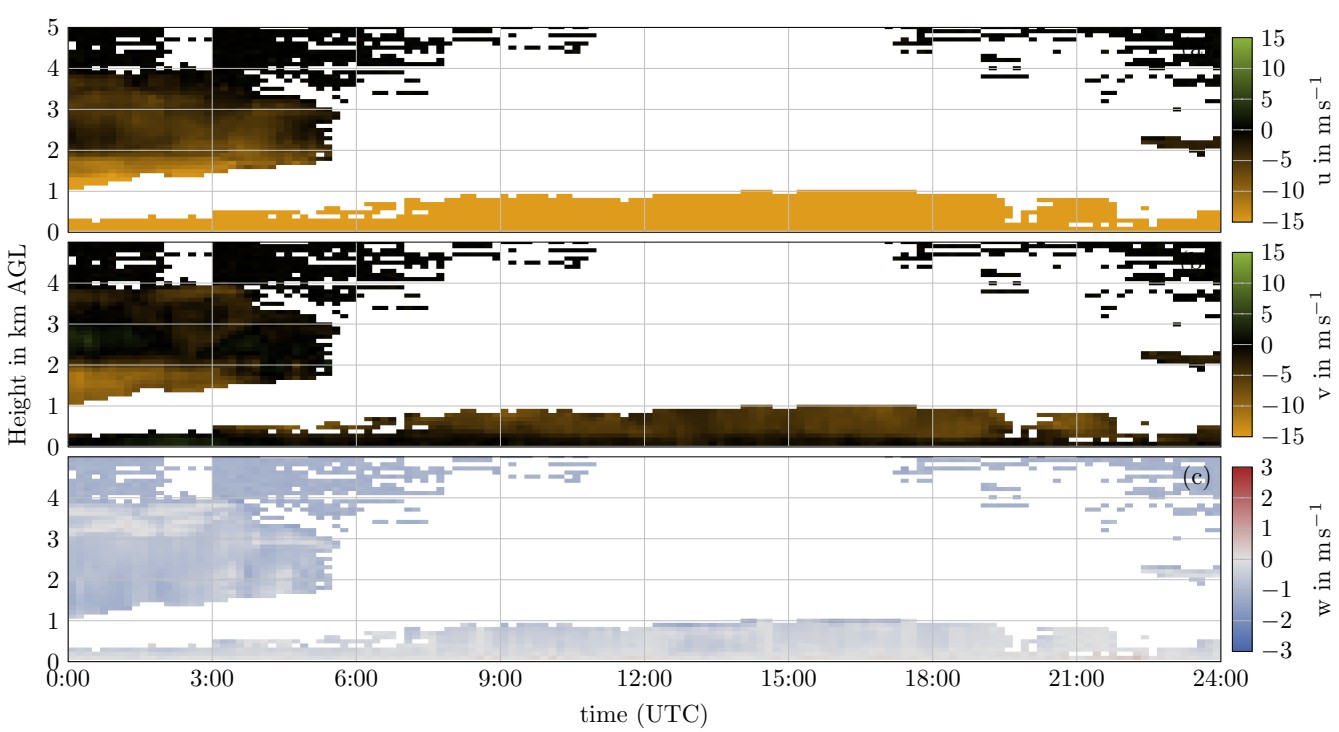

**Figure D1.** Result of the standard module chain applied for the Streamline XR+ at Neumayer Station on 19$^{th}$ May 2024. A number of wind vectors indicate a vertical wind of $-1\,\mathrm{m\,s^{-1}}$ without horizontal wind. The reason is a peak in the radial velocity noise distribution around $-1\,\mathrm{m\,s^{-1}}$.

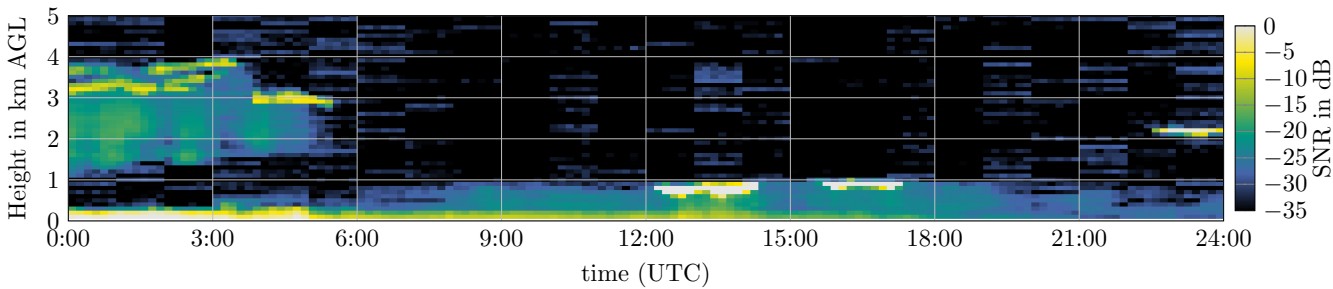

**Figure D2.** Doppler lidar SNR at Neumayer Station on 19$^{th}$ May 2024. The Median of the SNR values of all measurements available per bin is displayed.

The StreamLine XR+ at Neumayer Station exhibits peculiar, temporally and vertically extensive regions with vertical winds of approx. $-1\,\mathrm{m\,s^{-1}}$, without apparent background wind (i.e. horizontal wind speeds of approx. $0\,\mathrm{m\,s^{-1}}$). Fig. D1 shows





an example of this effect. The corresponding SNR is displayed in Fig. D2. While some of the retrievals could be physical,
e.g. due to snowfall in quiescent air, the continuity and frequent occurrence of the vertical winds without horizontal winds
is implausible. Closer analysis reveals that the vertical velocities are attributable to frequent radial velocity measurements
around $-1\,\mathrm{m\,s^{-1}}$ during weak SNR conditions below $-22\,\mathrm{dB}$ (Fig. D3). Since the $-1\,\mathrm{m\,s^{-1}}$ radial velocity measurements are
independent of the scanning direction (i.e. also azimuth direction), no horizontal winds and solely vertical winds are retrieved,
if the $-1\,\mathrm{m\,s^{-1}}$ measurements present the overwhelming majority of measurements.

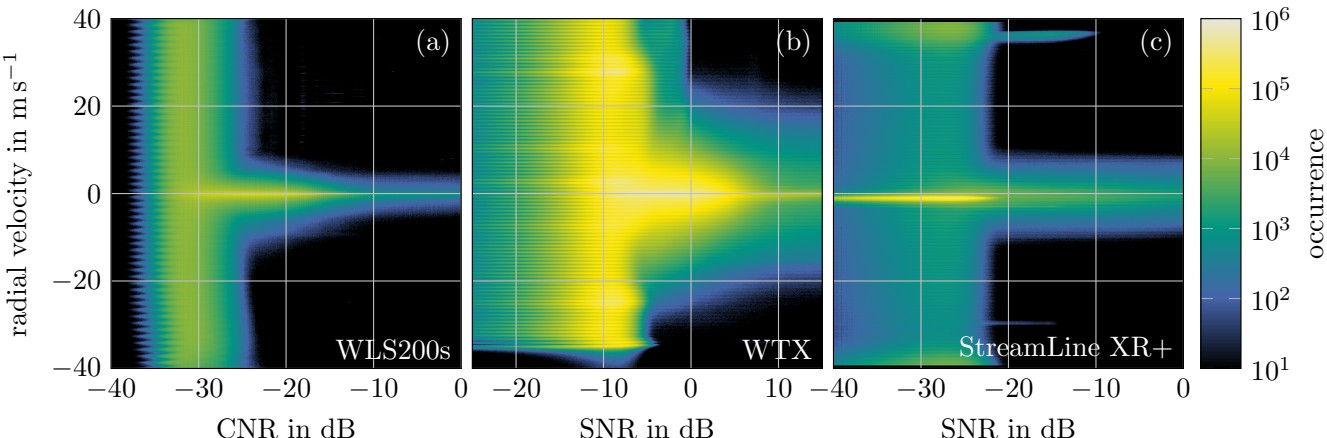

**Figure D3.** Occurrence of radial velocities for different SNR values for WLS200s at Payerne (a), WTX at Villingen-Schwenningen (b), and
StreamLine XR+ at Neumayer Station (c). All radial velocities are summed up in bins of $0.1\,\mathrm{dB}$ and $0.1\,\mathrm{m\,s^{-1}}$ between June and August
2023 for (a) and (b), and between $3^{\mathrm{rd}}$ January and $26^{\mathrm{th}}$ November 2024 for (c). The instruments and locations correspond to the sites used for
the validation using radiosondes (Sect. 5). Detectable issues might be connected to the single instrument, the settings, or local circumstances
and cannot generally be attributed to the respective Doppler lidar type.

The non-uniform distribution of radial velocity measurements during weak signal conditions is traceable to a previously un-
reported issue of the StreamLine XR+ at Neumayer Station. Likely, internal system issues (e.g. electro-magnetic interference of
hardware components) contaminate the spectrum used for radial velocity estimation (Schmithüsen, Kayser, Bühl, Engelmann,
Radenz 2024, personal communication). Since the raw spectrum data is not available, the radial velocity estimation cannot be
improved through an improved background noise correction and subsequent reprocessing in this study.

Nevertheless, StreamLine XR+ wind profiles from Neumayer Station are included in the radiosonde comparison since they
serve to show both a limitation and an advantage of the presented modular wind profile retrieval software: First, the software
should not be used to process Doppler lidar data irrespective of data quality, without the operator's reflection on the provided
results. The consensus approach used in the least squares fit retrieval and the background wind estimation rely on a broad
distribution of radial velocity noise. In the Neumayer case, only user experience and meteorological reasoning allow to judge
the frequent $-1\,\mathrm{m\,s^{-1}}$ radial velocity measurements as implausible, a task which cannot be provided by the software. Second, if
an issue with the data is identified, the modular software can be easily adapted to deal with the problem. In the Neumayer case,



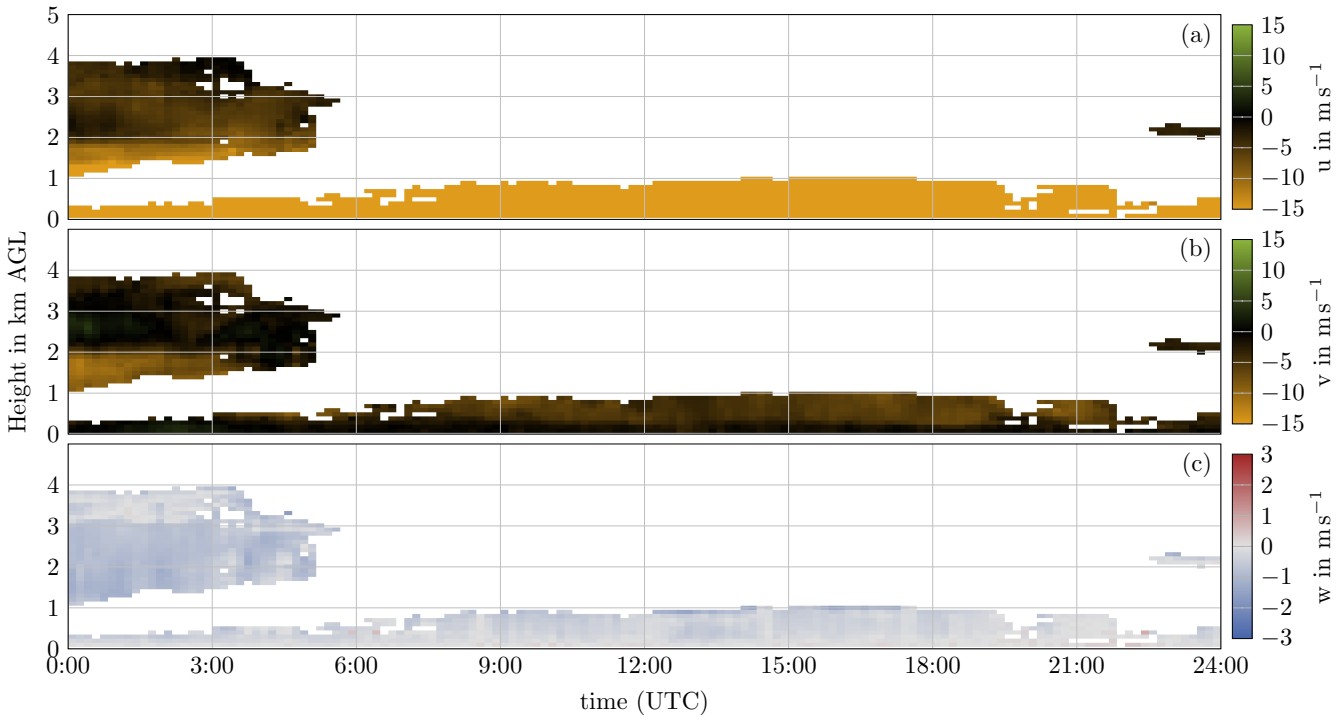

**Figure D4.** Result of the extended standard module chain applied for the Streamline XR+ at Neumayer Station on 19$^{\text{th}}$ May 2024. Radial velocities between $-2\,\mathrm{m\,s^{-1}}$ and $-0.35\,\mathrm{m\,s^{-1}}$ are excluded from the retrieval. Most probably disturbing wind vectors appeared in Fig. D1 are rejected.

an additional level 1 consideration flag is introduced in the module chain: radial velocities between $-2\,\mathrm{m\,s^{-1}}$ and $-0.35\,\mathrm{m\,s^{-1}}$ are excluded if they exhibit an SNR below $-22\,\mathrm{dB}$. This filter prevents ingestion of the contaminated radial velocities prior to wind vector retrieval, but still allows for retrieval even in weak SNR conditions with the remaining measurements.





**Appendix E: Supplementary figures**

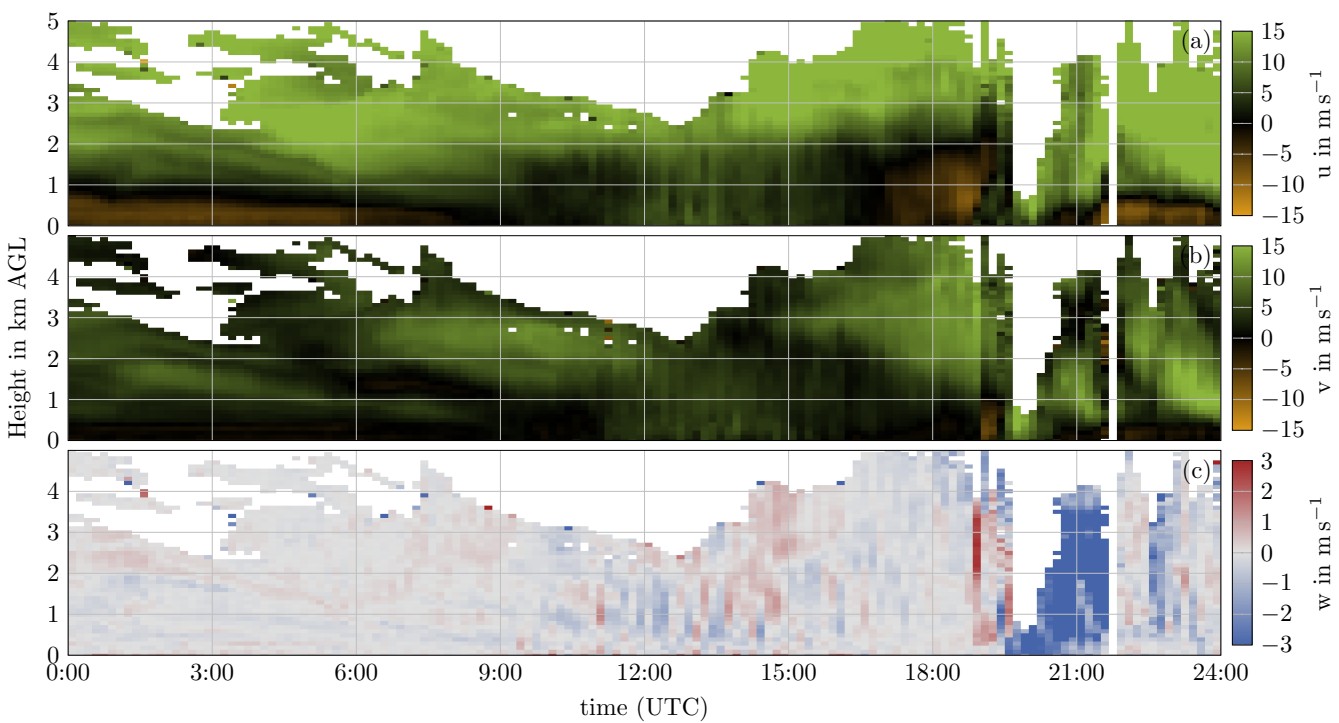

**Figure E1.** Simple retrieval considering measurements with a CNR threshold of $-30\,\mathrm{dB}$. As no confidence background is applied, transient conditions (at 20:00 UTC in particular) are covered better than in Fig. 10. However, some erroneous wind vector retrievals are also incorporated.



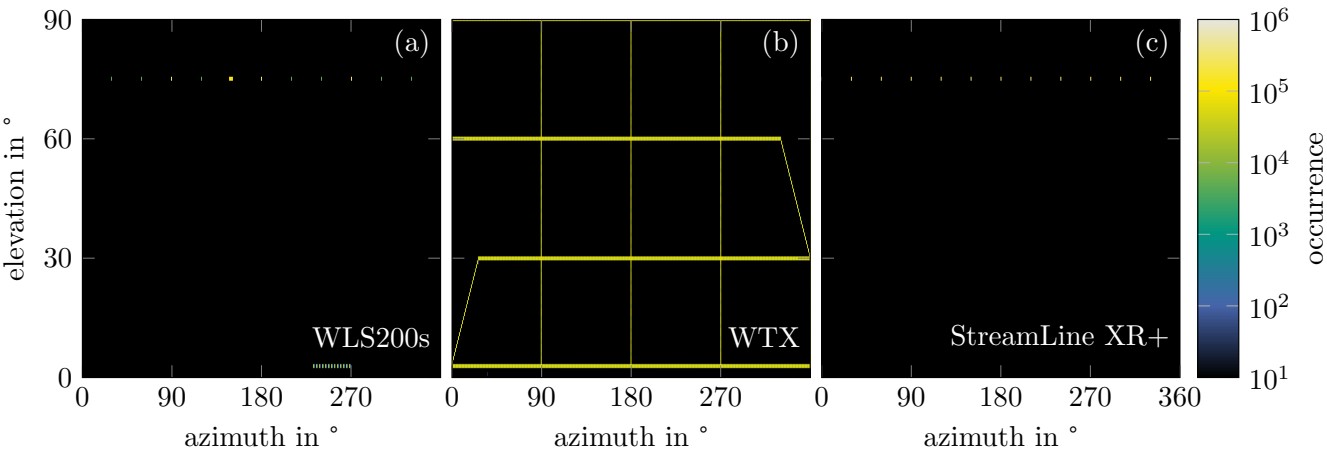

**Figure E2.** Occurrence of azimuth and elevation scan angles for WLS200s at Payerne (a), WTX at Villingen-Schwenningen (b), and Stream-Line XR+ at Neumayer Station (c). All angles are summed up in bins of 1° between June and August 2023 for (a) and (b), and between 3rd January and 26th November 2024 for (c).

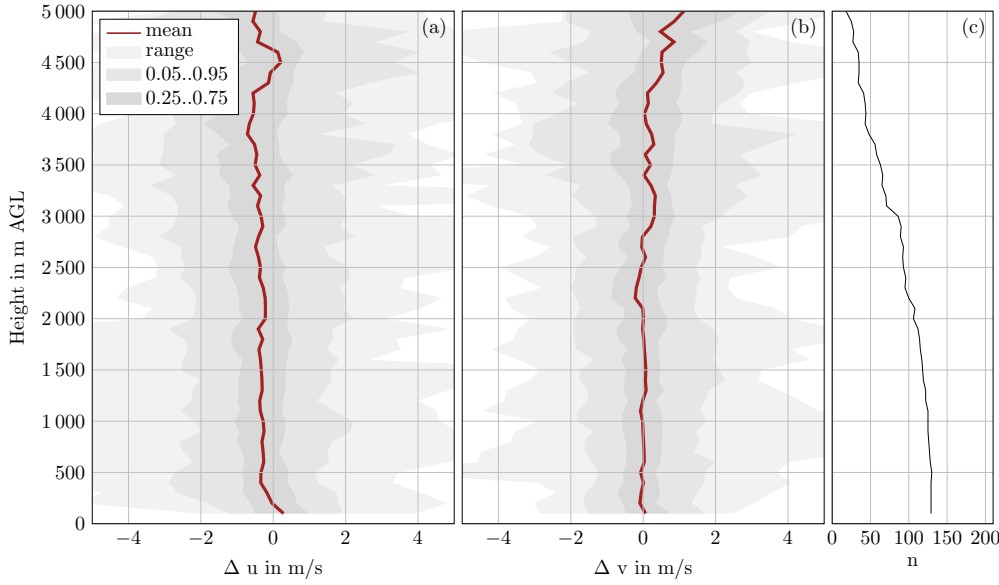

**Figure E3.** Horizontal wind component difference between radiosondes and the WTX retrieval at Villingen-Schwenningen. The same situation as in Fig. 14 is shown, however, at an increased temporal resolution of 5 min of the lidar retrieval.





*Author contributions.* Anselm Erdmann: conceptualization, methodology, software, validation, writing – original draft, visualization. Philipp Gasch: conceptualization, methodology, validation, writing – original draft.

*Competing interests.* The authors declare that they have no conflict of interest.

*Acknowledgements.* The authors thankfully acknowledge the continuous support of Maxime Hervo (MeteoSwiss) for providing Doppler
lidar and radiosonde measurements at the Payerne, Grenchen, Schaffhausen and Zürich Stations. The authors also thankfully acknowledge the support of Holger Schmithüsen (Alfred Wegener Institute, Helmholtz Centre for Polar and Marine Research) for discussions about the Doppler lidar at Neumayer Station and for providing radiosonde measurements from Neumayer Station. The authors also thank Markus Kayser, Johannes Bühl, Ronny Engelmann, and Martin Radenz for sharing their experiences and insights into the noise characteristics of the Halo Photonics StreamLine XR+ Doppler lidar. Last but not least the authors thankfully acknowledge the support of the Swabian MOSES
measurement campaign team in conducting the field measurement campaign. Especially the essential effort of the KITcube team under the coordination of Andreas Wieser is thankfully acknowledged.



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
