# Peer review of "A modular wind profile retrieval software for heterogeneous Doppler lidar measurements"

_Geoscientific Model Development, 2024_

## Author Comment (AC2)

**Replies to referee comment 1**

*The paper describes a software package to retrieve wind profile in the atmosphere from Doppler wind lidar. The paper is of good quality and describes well what the basic concept of wind retrieval from Doppler wind lidar are, and how this software implements them. The software uses a clearly defined data structure and is of modular design which allows to flexible set up retrieval chains. The presented examples contain interesting approaches to cope with noisy data, and to either identify reliable data points or reject them. The description is for the most parts clearly written and well understandable.*

We want to thank the referee for the detailed evaluation, positive feedback, valuable criticism, and many insightful comments and suggestions for improvement.

*Nevertheless I have some comments:*

*Sampling in height bins:*
*The retrieval takes all Doppler speeds measured within one height bin, If I understood it right they are used as if all of them were measured at the same height. This neglects that the wind speeds especially in the boundary layer show a strong non-linear height dependence. Basic assumption of the wind retrieval is that the radial Doppler wind speeds measured in the different tilted beams are all result of the same wind vector. By bringing together different heights in one retrieval this assumption is violated. In my opinion it would be necessary to first interpolate the radial velocities to one single height within every height bin. The authors should discuss this in the text.*

The extent of the error due to not interpolated radial velocities depends on the vertical resolution. It is important to select a suitable vertical resolution, which typically varies between 25-100 m (for systems presented in this study). We added the discussion on the vertical resolution to the manuscript (see detailed comment). We additionally developed a level 1 module which is implements the radial velocity interpolation as suggested. We applied it for the example shown in the manuscript (Fischerbach, 11[th] July 2023). Figure 1 shows the result of the standard module chain (without radial velocity interpolation). Figure 2 shows the result with interpolation of the radial velocities. The differences between Figure 1 and Figure 2 are displayed in Figure 3.

[Figure]

Figure 1: Result of the standard module chain without interpolation of the radial velocities (Fischerbach, 11[th] July 2023).

The selected day covers different meteorological conditions. Noticeable differences are observed in edge areas, as well as at mid-levels between 9:30 and 17:00, and after 19:00. The variance of the radial velocity residuals is shown in Figure 4. The variance is high where high differences are present in Figure 3, showing the impact of turbulence causing differences between measurements from neighbouring range gates. In case of systematic error due to vertical interpolation, deviations would be expected also in bins with a low variance. In contrast,

[Figure]

Figure 2: Result of the standard module chain with additional interpolation of radial velocities. The other settings are the same as for Figure 1.

the variance is low from 4:00 to 9:00 between 1 km and 3 km. The differences are small during this time in these ranges, despite the presence of vertical wind shear. Hence, we do not expect a significant systematic error if no vertical radial velocity interpolation is performed prior to the retrieval.

Generally, the retrieved wind vector represents the complete bin in the presented retrieval. In the case of high variance within the retrieval volume, there may be a difference to the wind vector at the bin centre. For the selected configuration of Doppler lidar and retrieval settings, we see an advantage of having additional radial velocity measurements when multiple ranges are mapped to the same height bin. However, we recognize, that there could be configurations where just one range is mapped to a bin, or where the wind vector is needed only for specific heights, but not for complete bins. Hence, we added a calculation module interpolating radial velocities according to the vertical resolution settings to the repository. This module can be added as first module to the module chain if using the interpolation is on advantage. This module (Variables_height_interpolation_l1) is described in the manual (currently Section 4.2.14).

We adapted the manuscript to reflect this discussion and offer the additional interpolation module for users who would like to interpolate the measurements before retrieval.

*Combination of different scans patterns:*
*The common way for wind profile retrieval is to perform a certain scan pattern which should evenly cover the measuring volume (see [Päschke et al., 2015], mentioned in the text). Different scan patterns can be used to address different questions (fast scans like DBS with only few beams to achieve high temporal resolution, PPI (or VAD) scans with many beams to reduce the uncertainty within one scan, RHI scans to eventually resolve spatial variability at least in one vertical plane, etc.). Several scans of the same kind can be combined by averaging.*
*Here is a different strategy proposed which combines all in a certain time interval available scans (hence the term 'heterogeneous measurements'). This introduces asymmetry in the spatial coverage and may lead to larger errors as is shortly mentioned in section 4.1 when describing the standard module chain and the removal of data from low elevation beams at large distances (line 375). I recommend to emphasise stronger this strategy but also discuss the effects and dangers of an uneven coverage of the measuring volume in section 2.*

We restructured this section as suggested in the detailed comments. In the revised version, we introduce the PPI/VAD pattern as common pattern for wind profile retrieval, followed by patterns addressing the different interests you mentioned. An asymmetry is generally possible and sometimes desired by individual users (e.g. [Wang et al., 2015, Wang et al., 2016]). We observed asymmetric scan geometries for Payerne (Fig. E2 (a)), where a sector scan is performed around 150° azimuth at 75° elevation (the shallow sector scan around 270° azimuth is below the elevation angle threshold). To prevent wind vector calculations from highly asymmetric measurements, the CN limit is used. We emphasized the issue of asymmetric scan patterns in this context in the revised manuscript. The final decision on the applied scan pattern lies with the lidar operator and the

[Figure]

Figure 3: Differences between Fig. 2 (positive) and Fig. 1 (negative).

[Figure]

Figure 4: Variance of the radial velocity residuals for Fig. 1.

customisable module chain gives opportunities for retrievals using only selected directions or scans.
We revised the manuscript to reflect this discussion and emphasize the importance of balanced scans.

*Extrapolation in regions with low CNR:*
*An important part of the advanced retrieval is the attempt to use even data at low CNR. To achieve this a first guess wind profile is derived and inter- and extrapolated to regions with low CNR. The kind of this extrapolation is not documented. The resulting fields are used in an iterative process as 'confidence background'. Calculated wind speeds deviating not too far from this confidence background are accepted even for low CNR values. I am wondering whether this extrapolation is robust, especially in upper heights, where CNR is low over large regions. The systematically large differences in the upper part of the profiles in fig's 13, 14 and 15 might be result of this extrapolation. The authors should discuss this in the text. It would be nice to have an analysis how the retrieval performs for low CNR and whether the difference to the radiosonde data for these low CNR regions is acceptable.*

Previous studies have also used the concept of background estimates for refined and extended retrieval availability at levels with low CNR and reduced data availability (e.g. [Baidar et al., 2023]). We performed an inpaint_biharmonic extrapolation. The extrapolation is implemented with a seamless image restoration method (`https://scikit-image.org/docs/stable/api/skimage.restoration.html#skimage.restoration.inpaint\_biharmonic`). We added explanations to the manuscript. For the standard module chain it is important to have an extrapolation method that is not relying on previous measurements or model data (like it is done in optimal estimation approaches). Hence, we resort to an established restoration method from image processing. If knowledge about the local circumstances allows also optimal estimation, a replacement of the image-based extrapolation module might improve the guess. However, this knowledge is not available in every case. To

prevent retrievals due to spurious measurements randomly in agreement with the confidence background we use the NREL threshold. NREL represents the fraction of valid measurements needed in comparison to all measurements in a retrieval bin. If noisy measurements agree with the extrapolated background by chance, the fraction of measurements doing so can be expected to be small. To provide further control, we have now also limited the vertical and temporal extent of the background extrapolation, which can now be set by the user (maximum 1 km vertical, 1 hour temporal extrapolation for the standard module chain, see our answer in the minor comments section below). In addition, we now investigate the stability of the retrieval at low CNR, also suggested by referee 2, we have performed the CNR analysis as desired. The $\Delta u$ and $\Delta v$ deviations in comparison to the radiosonde are shown as a function of CNR in Figure **??** (Figure 16 in the revised manuscript). The median CNR value of all measurements considered for the wind vector calculations is used. There is no strong increase of error levels for lower CNR values detectable. A reason for the small increase observed could also be the distance between lidar and radiosonde, which is cross-correlated with CNR. This is discussed with additional Figures (E4, E5) in the revised manuscript. CNR typically decreases with altitude where the average radiosonde to lidar distance increases.

We added more details on the extrapolation to the manuscript. We introduced a range/time limit for the extrapolation. We investigated the performance for low CNR values in the radiosonde validation section.

*Description of basic scan patterns:*
*As the paper also addresses researchers not familiar with all the details of doppler Lidar retrieval it would be beneficial to give a compact, and clear overview of possible scan patterns, sources of errors etc. The word scanning is used in an ambiguous way, the step-stare scan mode is mentioned but continuous scan mode not.*
*To retrieve the 3D wind vector it is necessary to tilt several beams in different directions - this is called a scan. Common scan patterns are PPI (or VAD), RHI, etc. Scans can be performed in two different modes: step-stare and continuous. There is a trade-off between these modes (varying orientation, dead time during movement, time for one scan, etc.).*
We agree that the previous description of the term scanning and the scan pattern should be improved and extended, and now distinguish between continuous and step-and-stare scans. In addition, we have now revised the introduction of the basic scan pattern available to lidar operators.
We changed the introduction and removed the ambiguous wording: first we introduce the differences between the step-stare and the continuous mode. Subsequently, patterns in the step-stare mode and patterns in the continuous mode are introduced and trade-offs are discussed.

*There are two main sources of error:*
*the doppler retrieval has always some uncertainty, which increase with decreasing CNR (or SNR) (hence the CNR thresholding as filter strategy) and the violation of the homogenous flow assumption due to the separation of the tilted beams by e.g. turbulence. The along beam doppler uncertainty maps, depending on the elevation angle, to the retrieved wind components. With increasing elevation its contribution to the error of the horizontal wind component increases. With decreasing elevation the separation between the beams becomes larger and the error due to turbulence becomes larger.*
We also agree with the sources of error in Doppler lidar wind profiles, besides other instrument or operator errors (e.g. erroneous orientation of the lidar, second-trip echoes due to high pulse repetition frequencies). LES-based simulator studies have shown that the error due to turbulence decreases for more shallow elevation angles (closer to the horizontal), at the cost of an increased measurement footprint (e.g. [Gasch et al., 2020, Gasch et al., 2023, Rahlves et al., 2022, Robey and Lundquist, 2022]). The decrease is due to the horizontal components being observed more directly, resulting in a reduced impact of the vertical wind fluctuations in the retrieval.
The sources of error are discussed in more depth now.

*Range of biases*
*Figures 13, 14 and 15 show the bias between wind speed profiles retrieved with this method and radiosondes. While it is nice that on average the bias is close to zero the percentiles show that still 10% of the wind speeds deviate by more than about 4m/s from the radiosonde data. This could be compared with the results of Rahlves et al. (2022) and Robey and Lundquist(2022) and discussed.*
Besides the instrument and retrieval errors, another cause of deviations in the radiosonde intercomparison are spatial and temporal mismatches in the atmospheric volume probed by the two systems. Doppler lidars probe atmospheric volumes and radiosondes are advected with the wind and sample one-dimensional profiles, increasing in distance from the Doppler lidar with altitude. These sampling difference are unfortunately unavoidable. In contrast, the two studies mentioned evaluate the Doppler lidar measurements against reference truths taken from the LES at the location of the lidar measurements. Hence, a direct comparison with the LES-based results from the two studies is difficult. Additionally, our real-world comparison includes a variety of meteorological

conditions with differing degrees of boundary layer turbulence, flow heterogeneity as well as mesoscale disturbances. This further complicates the situation. While we acknowledge that a real-world evaluation of the results presented in the two studies would be useful, we think it is beyond the scope of this study. We now discuss the potential sources of error in the radiosonde-lidar comparison in more detail. We also included additional analysis on the effect of decreasing CNR and increasing distance in the radiosonde-lidar comparison to investigate potential retrieval and sampling effects.

We adapated the discussion on potential errors in Doppler lidar retrieved wind profiles. We discuss potential effects in the radiosonde lidar comparison in more detail. We added two plots showing the error as a function of the radiosonde distance and position.

- - - - - - - - - - - - - - - - - - - - - - - - - - - - - - -

*Detailed comments:*

*Line 64: period is missing*
Thank you for the careful reading.
Changed.

*line 94: ambiguous use of the word 'scanning' , especially with the list of scans to follow. Are here meant general measurements in different directions or especially measurements during continuous movement? An explanation and discussion of step-stare and continuous mode is missing.*
We agree and have adapted the description in-line with the above discussion. What we meant by scanning is to move the scanner head in different directions.
We have adapted the section in line with the above discussion on scanning.

*line 97: alternative name for PPI is VAD from 'Velocity Azimuth Display' (doppler velocities are 'displayed' as a function of azimuth. Wind components are retrieved by fitting a sine curve, |v| is proportional to the amplitude etc ...) already cited in Browning and Wexler (1968), stems from radar. Both terms are not perfect: PPI in radar technique means a scan at very low elevation and its presentation on a screen, VAD can also refer to the display of wind vectors as a function of height and time sometimes used in aviation.*
We agree, that there is a non-ideal and non-standardized usage in literature. We prefer to call the retrieval technique as 'VAD', but others also apply the same term to a constant elevation scan pattern. We prefer to call constant elevation scans 'PPI', to avoid confusion with the retrieval technique.
We have added a line stating that 'PPI's are also termed 'VAD' alongside a reference to the original study by [Browning and Wexler, 1968]

*line 99: With one RHI at one azimuth it is not possible to retrieve a wind vector, it should be mentioned that this is only possible with a combination of RHIs in different azimuth directions.*
Yes, we adapted the discussion to make this unmistakable.
We have adapted the section in line with the above discussion on scanning.

*line 100: "arrangements of fixed-direction stares, e.g. in step-and-stare"*
*This gives the impression that the above described scan patterns are not done in step-stare mode but instead in a different way (which is never named in the paper). I would recommend to mention first the different modes (step stare and continuous), discuss advantages and disadvantages and then list the principal scan patterns.*
We agree, this was not well explained.
Following your suggestion we have adapted the section in line with the above discussion on scanning.

*line 104: "RHI scans allow for measurements closer to the ground,"*
*A PPI can also be performed at lower elevations. I do not see why it must be a RHI,*
Correct, thank you for pointing out this error.
We have adapted the section in line with the above discussion on scanning.

*line 111: "The wind vector retrieval error is determined by ..."*
*I do not understand, is this doubled in the next sentence ?*
We have reworded the sentence to be more clear and avoid doubling of information.
Adapted to: "*The wind vector retrieval error is determined by three main factors: First, by the applicability of the wind field assumed in the retrieval; second by lidar measurement errors, e.g. due to random radial velocity fluctuations or beam pointing errors; and third by numerical errors during the calculation.*"

*line 117: "Steep measurements"*

*I guess you mean "... at high elevation angles"*
Yes.
Reworded.

*line 118: "shallow scans"*
*I guess you mean "... at low elevation angles"*
Yes.
Reworded.

*line 118: "... retrieval errors due to turbulence typically average out within a 10 min to 30 min span,"*
*But the uncertainty never vanishes - you cite Rahfles et al (2022) who showed that RMSD becomes never zero and is surprisingly large.*
Correct.
We adapted the statement: "*If appropriate scan elevation angles are used, retrieval errors due to turbulence typically become negligible within a 10 min to 30 min span, depending on atmospheric conditions, but do not vanish completely.*"

*line 125: "... some instruments provide a signal-to-noise ratio (SNR),"*
*I would appreciate some words what the difference between CNR and SNR in Windlidardata is, and whether there is a difference in the use of it.*
This is a good point. Unfortunately, to our knowledge, the difference between CNR vs. SNR is not well documented in lidar literature. Our understanding is that SNR describes the ratio of the signal peak height to the surrounding noise floor. Narrow- vs. wide-band SNR estimates exist depending on the width of the noise floor considered. CNR on the other hand describes the ratio of the areas below both spectral curves (signal vs. noise), making CNR more independent of changes in the signal curve, e.g. due to spectral broadening by turbulence. Unfortunately, the only references available to us for this are manufacturer internal documentations, which we are not allowed to distribute further. Since we are not aware of studies documenting the differences mathematically or methodically, we would like to refrain from going into details in our manuscript to avoid establishing incorrect definitions.
No changes.

*line 123: "Filtering with the carrier-to-noise ratio (CNR) is ..."*
*Please mention that with decreasing CNR the uncertainty/the error increases, Also mention parameterizations: There are estimates for the doppler retrieval uncertainty - see eg Pearson and Collier (1999, doi:10.1002/qj.49712555918), Manninnen et al 2016 (doi:10.5194/amt-9-817-2016) who they refer to Rye and Hardesty (1993) and O'Connor et al. (2010),*
*Also worth mentioning is that if CNR becomes too low the doppler speed becomes evenly distributed in the bandwidth (as is visible in Fig D3 in this paper).*
We have reworded the section to give more details.
Modified text, included references.

*line 127: "Potential causes of erroneous measurements despite high CNR..."*
*I am missing precipiation ...*
We've included precipitation among the flying objects.
Added precipitation.

*line 138: "if the beam directions of the radial velocity measurements are not sufficiently dispersed ..."*
*I would prefer the word 'distributed [in space]' instead of dispersed. (the term dispersion has a different meaning in optics, and another in Chemistry, ...)*
Ok, we've adapted the wording.
Modified the wording.

*line 140: "The condition number (CN) is a measure for the robustness of the beam dispersion"*
*I would write: "... CN is a measure for the robustness of the equation system with respect to errors in the input variables. "*
Thank you for the suggestion, we've adapted the wording
Modified sentence.

*line 143: "Maximizing the number of considered measurements allows for the smallest assumption violation, due to the beneficial effect of averaging."*
*This is the basic argument for the combination of several different scan patterns in the retrieval. In this uni-*

*versal formulation I doubt it, especially because it could be a violation of the argument about the spatial beam distribution of the previous paragraph. Fig.1 is a nice example: the RHI has a much higher spatial coverage than the VAD, but in only one direction. In a VAD scan a higher number of measurements in one direction leads to a lower condition number (Paeschkae et al 2015).*

We have modified the statement to make it less general.

Modified sentence.

*line 163: "Changing absolute range gate heights for different laser beam positions can be overcome by binning the measurements with respect to the height above ground."*

*If understand right this means that radial velocity measurements from different heights within one height bin are assumed to represent the same wind vector. This neglects the height dependence of the wind, and accordingly violates the homogenous wind field assumptions.*

We refer to the effect of vertical interpolation discussed above. Of course, the vertical binning should occur in a meaningful way, i.e. be related to the average lidar pulse length/radial velocity resolution. Typical values are below 100 m for lidars, based on the above discussion we think that vertical changes in the horizontal can mostly be captured except when very close to the ground.

We've added a statement reflecting the above discussion.

*line 180: "timestamps of the measurements,"*

*Any requirement whether this should be the begin or end of the averaging interval?*

We conduct the binning individually for every measurement. No principal requirement on the time stamp exists, but of course the user should be aware how their system labels the measurement time. For Doppler lidars indicating the begin or end of the interval, the timestamp needs to be adapted as part of the level 0 to level 1 conversion in case the measurement duration is not negligible.

We added statements indicating the typical scenario.

*line 181: "The time coordinate can differ between instruments at level 0 and will then also differ at level 1."*

*Unclear, Please clarify.*

We modified the wording, also requested by referee 2.

Adapted the statement for clarity.

*line 194: "... azimuth (time), elevation (time) ..."*

*When performed in continuous mode azimuth and elevation vary during the integration time. It should be mentioned which angles must be provided here (begin, middle, end of integration time).*

For our measurements, the centre value was used, however, this can in principle vary between instruments.

Modified.

*line 212: "Level 2 represents wind vectors ..."*

*I guess you mean 'Data level 2 repesents ..."*

Yes, thank you for the careful reading.

Modified.

*line 239: "flagging of precipitation."*

*Is there a module to flag precipitation? If yes: how is this done ?*

Currently, no precipitation flagging module exists, hence we omitted this statement.

Reworded.

*line 250: "The validity_probability variable contains an acceptance (1.0) or rejection (0.0) value for each bin according to a user-defined threshold."*

*That seems rather to be a flag, or is it possible to implement fuzzy logic based on values between 0 and 1. ?*

In principle this flag can be probabilistic with values between 0 and 1. However, the current standard module chain only uses 0 or 1 as values, i.e. flagging. Since probabilistic values are possible in principle we would like to keep the statement as is.

No changes.

*Table 1: "(8) Bin_statistics_l1_to_l2 ... provide CNR and Doppler spectrum width information to l2"*

*Interesting! How is this done ?*

*Doppler spectral width can be regarded as a standard deviation - not in time but in the measuring volume of the respective range gate. In Sathe et al (2015) equation 4 can be seen that the along beam variance depends on the variances and covariances of the principal components of the wind vectors. Thus a retrieval of doppler spectrum*

*width in the principal spatial components could be done with this. But does it work ?*

The short labelling is maybe misleading. The module 'Bin_statistics_l1_to_l2' compiles statistics for all l1 measurements that fall into the same l2 bin. These statistics are then an additional variable for l2, containing the corresponding value for each bin. Currently, mean, median, standard deviation, variance, and count are available as mathematical operators for the computation. The current per bin statistics is not calculated with respect to the individual measurement directions (as in [Sathe et al., 2015]), although this would in principle possible since the directional information is also available in the l1 dataset. We will look into the feasibility of such a directional calculation in the future. In the meantime, the documentation on how to implement additional modules is available in the software documentation.

No changes.

*line 376: "Such an imbalance negatively impacts the vertical wind vector calculation and is, therefore, undesirable."*
*This is the argument against the general statement from line 143 that the number of measurements should be used. Line 143 should be adapted. I am wondering how this affects only / especially(?) the vertical wind. Maybe that is result of the strong wind shear in the height bins close to the surface?*

You're correct. We have reworded the old statement in line 143 to account for the fact that balanced scans are optimal. In principle, an imbalance can also occur in other components if they are not well represented. However, we've frequently seen this imbalance for the vertical wind component if predominantly very shallow PPI scans are used for retrieval. This is the reason why we have implemented the additional module to exclude specific elevation ranges, which could also be used for azimuthal filtering.

Kept the statement as is since the preceding discussion was modified accordingly.

*line 384: "... a strict CNR filter (the default threshold for Leosphere WLS200s is -25 dB) ... a weak CNR validity flag (default threshold -30 dB for Leosphere WLS200s)"*
*A guideline on how can these thresholds could be determined for instruments from other manufacturers would be desirable. I would be also of interest to mention at some point how these kind of thresholds or parameters are provided to the software and how they can be changed.*

The thresholds and how they are obtained are now discussed in more detail in the CNR section. We provide thresholds for three commonly used systems (WLS, Halo, Lockheed Martin WindTracer). The way to modify them (by changing the settings (*.ini) file) is documented in the user documentation of the software, which we now mention in the text. We now also provide a more in-depth discussion of the reasoning behind the thresholds in-line with requests by referee comment 2.

We added this information.

*line 392: "... the volume enclosed in the convex hull spanned by the unit vectors"*
*This is unclear: If I imagine a VAD scan with constant elevation the volume spanned by the endpoints is a polygon (lets say a disk) with vertical extension equal to zero. Its volume would be zero. If it is the whole cone spanned by the vectors it is easily much larger than the threshold of 0.2.*
*Please clarify.*

It should have been the 'whole' cone. The unit vectors start at (0,0,0). You're statement is correct for a 0° elevation PPI, which is a scan that will be excluded by the filter. The default threshold is 0.042, as stated in the manuscript. You're correct, this volume threshold is easily exceeded and only serves to exclude bins with measurements from very limited sectors, e.g. only scans with elevation angles above 75°. As any parameter, this parameter can be adjusted by the user if desired or needed.

Clarified the meaning.

*line 406: "... even at low CNR reliable measurements may be available in some circumstances,"*
*Radial velocity uncertainty increases with decreasing CNR. Values will lie in a large interval around the true value. The value can lie just by chance in the trustworthy range around the 'confidence background'. It does not contain information. It becomes trustworthy just by coincidence. I see this is addressed by limiting the share of accepted values to a minimum.*

Correct, avoiding noisy measurements falling into the confidence interval by chance is the reason for requiring a fraction of valid measurements compared to the overall number of measurements.

No changes.

*line 412: "...the confidence background is extrapolated in module (7.2)."*
*How does this extrapolation work ?*

We performed an inpaint_biharmonic extrapolation, see our above answer in the 'major comments' section ('Extrapolation in regions with low CNR').

We added the information, an extended discussion and extrapolation limits in the revised manuscript.

*In fig 9 it is obvious that wind speeds are extrapolated into large regions where the first guess gave no data. This bears the danger to create unrealistic values. Application of this confidence background to low CNR values may let pass through otherwise unplausible values.*

*Please discusss this.*

To avoid unphysical extrapolation in distant regions, we added an extrapolation limit to the extrapolation module in the revised version. In the default case, the limit is 1 km vertical extrapolation and 1 h temporal extrapolation. Additionally, as discussed above, we require a fraction of valid measurements compared to the overall number of measurements for a valid retrieval. If the share of radial velocity measurements within the tolerance is above a threshold (default: 0.2), the wind vector is retrieved from all measurements within the tolerance. In case of an implausible guess, the share of radial velocities supporting the implausible guess is assumed to be below the threshold, if the noise is randomly distributed. Handling cases where the noise is not randomly distributed is discussed in Appendix D for the Neumayer Station Doppler lidar.

Due to the introduced extrapolation limit, all wind profiles have been recalculated and the plots updated. Some iteration advantage (Fig. 11) is shifted towards later iterations. For Neumayer Station with very weak CNR signals, the number of wind profiles available in the radiosonde comparison is reduced from 212 to 200.

In addition, we now also investigate the impact of the CNR level on retrieval accuracy in the radiosonde comparison (see above answer on 'Extrapolation in regions with low CNR').

See above.

*line 463: "Whether a higher availability from more iterations justifies the increased calculation effort can be decided by the user in the individual application case."*

*It would be great to see whether the additionally retrieved wind vectors are of good quality.*

For the quality of the additionally retrieved points see our above answer in the 'major comments' section ('Extrapolation in regions with low CNR'). We do not see a systematic increase of error with decreasing CNR in our results.

See above.

*line 475: "Heterogeneous scans of type DBS, step-and-stare, and sector PPIs are used for wind profile retrieval (Fig. E2 (a))."*

*I see the sector PPI at an elevation of only a few degrees in a sector of only 45deg width. Does this data contribute to the retrieval or is it excluded as it is too low and too far away?*

The lower PPI conducted by the Meteo Swiss WLS200s at Payerne is filtered by the minimum elevation threshold of 15° used in the standard module chain. There is another sector PPI at 75° elevation and about 150° azimuth which is not filtered.

We added an overview of the default values used in the standard module chain to the appendix.

*line 478: "The lidar conducted PPI scans ... (Fig. E2 (b))."*

*This looks as if the lidar performed its scan in continuous mode. Please clarify.*

Correct, the WTX WindTracer Doppler lidar operated in continuous scan mode, which we now added this information.

The information was added.

*line 486: "... also termed VAD pattern for Halo systems ..."*

*This term is not Halo specific. You can find it for wind lidar scans already in Browning and Wexler (1968). The synonyms PPI and VAD could be introduced with the scan patterns.*

We wanted to mention the differing use of the term VAD in the Halo systems, since it may be confusing for the reader. The Halo systems define the 12 point step-and-stare pattern conducted at Neumayer as VAD in their terminology (i.e. in the output data file), which we find misleading. In our understanding, the VAD term coined by Browning and Wexler [Browning and Wexler, 1968] describes the retrieval associated with a continuous scan across the full azimuthal range, not a 12 point step-and-stare pattern. We avoided the term VAD in general, as we observe an ambiguous use between the scan pattern, and the display and retrieval technique. Therefore, we use the PPI term for continuous scan across all azimuth, and now introduce the definitions in the general scan description overview.

We now omitted the sentence to avoid confusing the reader.

*line 492: "At weak SNR, very frequent vertical winds with 1ms-1"*

*You could refer to fig.D3(c) - it is clearly visible*

Thank you, we added this reference.

Added reference.

*line 507: "Fig. 12 shows the comparison of the radiosonde measurements with the corresponding retrieved horizontal wind speed".*
*How are radiosonde and wind profile data matched: The radiosonde needs about 16minutes for 1km or more than 1h15m for the 5km displayed. Is it compared to the wind profile at start time, one profile in between or to several lidar profiles according to closest time ?*

The Doppler lidar retrieval is compared to the wind vector associated with the time/height bin valid for the respective position of the radiosonde, i.e. the nearest neighbour in time. We specify that the radiosonde ascents with approx. 5 m/s vertical speed. This means about 20 s are needed for 100 m. For 5 km, about 16 min and 40 s are needed.

Added information on nearest neighbour matching.

*Fig 12:*
*Instead of repeating the scatter plot with different, difficult to depict colours, it would be helpful to see differences lidar - radiosonde as a function of the parameter of interest i.e. radiosonde distance, vertical velocity and eventually wind speed.*

Thank you for the hint. We observed that this figure was rendered differently in different PDF viewers. We hope it is now clearly visible in all PDF readers. Nevertheless, we added the differences as a function of CNR (in Section 5.3) and radiosonde distance (in the appendix).

We revised the figure and added two additional figures showing the differences as a function of CNR and distance.

*Fig 12 - caption:*
*"For Villingen-Schwenningen, one radiosonde profile is omitted"*
*I guess the crosses are the data from this profile.*

Correct, we added this information to the figure caption.

Added information.

*Fig 13:*
*Systematic negative Delta v above 4km:*
*I recognize that the interquartile range does not deviate as much as (arithmetic?) mean, i.e. only few extreme datapoints with Delta << -4m/s (light grey shading) are responsible for this negative Bias. N is also low, so this maybe only one single Profile ?*

Good observation. We agree, very few data points cause the negative bias. The comparison at altitudes above 4 km is conducted at the border region of comparability, where the number of comparison points is below 20 and average distances between lidar and radiosonde are largest. Due to the small sample size, we are not surprised by the influence of individual points on the comparison, and the meaning of the arithmetic mean and interquartile range should be treated with caution. We added plots of all single radiosondes to the dataset https://doi.org/10.5281/zenodo.14844887. The "extreme" radiosonde is the Payerne 23:00 UTC radiosonde on the 11[th] of July 2023. The wind direction differs significantly at a wind speed of more than $30\,\mathrm{m\,s^{-1}}$. At 4.7 km height, the v component of the lidar is $5.0\,\mathrm{m\,s^{-1}}$ while the v component of the radiosonde is $14.7\,\mathrm{m\,s^{-1}}$.

Nothing changed, since we already stated that "Larger differences are observed above 4 km, attributable to the insufficient sample size and large horizontal distances between the radiosondes and the Doppler lidar."

*I am wondering whether this extreme bias is result of the extrapolation of the confidence background. Similar deviation are visible in fig 14 and 15.*

See reply above. See detailed reply in the major section 'Extrapolation in regions with low CNR'. For further impressions on single radiosondes we added plots of all radiosonde comparisons to the dataset and the deviations in dependence on the geographical position in Figure E5.

We added plots of all single radiosondes to the dataset.

*line 534: "For Payerne and Villingen-Schwenningen, the comparison shows good agreement"*
*What about the negative bias at Villingen-Schwenningen between 0 and 3km ? Ah, i see this is discussed later.*

Yes.

No changes.

*line 539: "The slight variation in the number of retrieved wind vectors ..."*
*This is an aliasing effect. Heights bins are every 100m. Lets assume your range bins map to heights every 30m. As 100m and 30m do not have a common denominator you get alternating 3 and 4 measurements in one height bin. As several measurements are collected in every height bin, the variation becomes a multiple of 3 and 4, respectively.*

Your description is correct. However, we think calling it aliasing could also cause some confusion, since the folding of Doppler frequencies above the Nyquist frequency is also called 'aliasing'. Since this effect is different

from the effect we see here we would like to refrain from calling it 'aliasing'.
No changes.

*line 547: "(bias) in the u component of about 0.4ms-1" I see negative values in fig 14.*
Thank you for the careful reading. Yes the bias is $-0.4\,\mathrm{m\,s^{-1}}$ and we added the minus sign.
Added minus sign.

*line 550: "The u-component shows *a* systematically higher mean values,"*
*(Typo with the 'a' )*
Thank you for this correction. We removed the 'a'.
Removed typo.

*The bias in the plot is negative and I thought it is 'lidar - radiosonde'.*
Yes, the bias is $-0.4\,\mathrm{m\,s^{-1}}$, see correction above.
See above.

*I have the impression that at least the total range of the deltas is larger for the u than for the v component.*
Yes, this is an excellent observation. The large spread in $\Delta$u is related to the climatology of the comparison. Radiosondes from Villingen-Schwenningen were launched during intensive observation periods targeting the initiation of thunderstorms. Due to the targeted launches the wind came predominantly from west. Due to the larger magnitude of u, the absolute deviations are also larger.
Added sentence stating so.

**References**

[Baidar et al., 2023] Baidar, S., Wagner, T. J., Turner, D. D., and Brewer, W. A. (2023). Using optimal estimation to retrieve winds from velocity-azimuth display (VAD) scans by a Doppler lidar. Atmos. Meas. Tech., 16(15):3715–3726.

[Browning and Wexler, 1968] Browning, K. and Wexler, R. (1968). The determination of kinematic properties of a wind field using Doppler radar. J. Appl. Meteorol., 7(1):105–113.

[Gasch et al., 2023] Gasch, P., Kasic, J., Maas, O., and Wang, Z. (2023). Advancing airborne Doppler lidar wind profiling in turbulent boundary layer flow – an LES-based optimization of traditional scanning-beam versus novel fixed-beam measurement systems. Atmos. Meas. Tech., 16(22):5495–5523.

[Gasch et al., 2020] Gasch, P., Wieser, A., Lundquist, J. K., and Kalthoff, N. (2020). An LES-based airborne Doppler lidar simulator and its application to wind profiling in inhomogeneous flow conditions. Atmos. Meas. Tech., 13(3):1609–1631.

[Päschke et al., 2015] Päschke, E., Leinweber, R., and Lehmann, V. (2015). An assessment of the performance of a 1.5 µm doppler lidar for operational vertical wind profiling based on a 1-year trial. Atmospheric Measurement Techniques, 8(6):2251–2266.

[Rahlves et al., 2022] Rahlves, C., Beyrich, F., and Raasch, S. (2022). Scan strategies for wind profiling with Doppler lidar–an large-eddy simulation (LES)-based evaluation. Atmos. Meas. Tech., 15(9):2839–2856.

[Robey and Lundquist, 2022] Robey, R. and Lundquist, J. K. (2022). Behavior and mechanisms of Doppler wind lidar error in varying stability regimes. Atmos. Meas. Tech., 15(15):4585–4622.

[Sathe et al., 2015] Sathe, A., Mann, J., Vasiljevic, N., and Lea, G. (2015). A six-beam method to measure turbulence statistics using ground-based wind lidars. Atmos. Meas. Tech., 8(2):729–740.

[Wang et al., 2015] Wang, H., Barthelmie, R. J., Clifton, A., and Pryor, S. C. (2015). Wind measurements from arc scans with Doppler wind lidar. J. Atmos. Ocean. Technol., 32(11):2024–2040.

[Wang et al., 2016] Wang, H., Barthelmie, R. J., Doubrawa, P., and Pryor, S. C. (2016). Errors in radial velocity variance from Doppler wind lidar. Atmos. Meas. Tech., 9(8):4123–4139.

---

## Author Comment (AC3)

**Replies to referee comment 2**

We want to thank the referee for carefully reading our submission, valuable hints and a discussion based on deep knowledge in Doppler lidar evaluation.

*This paper describes a wind profile retrieval software called AtmoProKIT to retrieve wind profiles from Doppler Lidar measurements. This open-source retrieval code is modular with standardized data formats and configurable module chains to handle complexity related to different types of Doppler Lidars, and different types of scans (PPI, RHIs, step-stare) in order to ensure quality controlled, standardized wind profiles with traceable uncertainties. While availability of an open-source, standardized wind profile retrieval code for Doppler lidar would benefit the scientific community, I am not sure the paper fits the scope of Geoscientific Model Development journal. It is primarily focused on describing the publicly available code to retrieve wind profiles from measurements, and nothing to do with model development or validation. One could argue that adaptation of this code could result in uniform wind profiles for data assimilation and model validation.*

Producing an open-source, standardized wind profile retrieval code is exactly the purpose of this submission, with data assimilation and model validation being the inherent drivers of this development. The processing of the Swabian MOSES campaign wind profiles was indeed carried out for data assimilation. We highlight this purpose in the revised version and added a reference [Handwerker et al., 2025].

Developing a toolbox to bring measurements into models fits the scope of GMD from our understanding, as GMD is not limited to complete models and comprises also utility tools like toolboxes according to `https://www.geoscientific-model-development.net/about/manuscript_types.html`. Before submission, we also studied other publications doing so within GMD (e.g. `https://doi.org/10.5194/gmd-18-101-2025`, `https://doi.org/10.5194/gmd-15-8983-2022`, `https://doi.org/10.5194/gmd-15-7557-2022`, `https://doi.org/10.5194/gmd-13-6111-2020`).

We applied for the joint GMD/AMT Special issue 'Profiling the atmospheric boundary layer at a European scale' `https://gmd.copernicus.org/articles/special_issue400_1209.html`. We think our purpose of processing unified and traceable wind profiles from heterogeneous measurements for use in models fits the scope of GMD and this special issue in particular, since the focus of this work is code-development based rather than measurement technique based. As noted, obtaining wind profiles from Doppler lidar is a widely used technique in general, however, few generalized and open-source algorithms to do so exist.

To ensure the paper fits the scope of GMD, we declare it as a software toolbox with a geoscientific application with paper type *model description paper* according to the link above. Hence, we added the name and the version of the software to the title as required for such model description papers. We hope highlighting the algorithm development for easier, more traceable and reproducible measurement processing for usage in modeling and data assimilation, and the clear indication of a software toolbox according to the GMD scope, makes this contribution acceptable for consideration in the GMD special issue.

We reworded the manuscript to clarify the relevance for modelling and data assimilation. We now reference the name and version of the software in the title of the manuscript as required by GMD.

*The following comments need to be addressed before the paper is accepted for publication.*
*Major Comments:*

*Wind profile retrievals from heterogeneous Doppler lidar data: Heterogeneous Doppler lidar data from different scans have been used to retrieve wind profiles on a different vertical retrieval grid before (e.g. [Tucker et al., 2009, Bonin et al., 2017, Pichugina et al., 2019] ). [Pichugina et al., 2019] used wind profiles retrieved at high vertical resolution near the surface from multiple PPIs, including very shallow elevation angles, to evaluate NWP for wind energy applications. Thus, the use of heterogeneous Doppler lidar data and the concept of retrieval volume is not novel.*

It is correct that height binning is not a novelty. In the original submission we mentioned that this binning concept is common for Doppler lidars operated on airplanes and ships. In the revised version, we clarified that this concept is an established concept to overcome the issue of range gate based retrievals and added the references you mentioned. We think the concept is useful and therefore utilize it to enable flexibility for retrieving wind profiles from arbitrary scans. However, the focus of the manuscript is on the open-source code development, ensuring standardization and reproducibility of procedures, not on the the development of new retrieval techniques.

We modified the discussion to clarify the scope and novelty of this study and included the references provided to give appropriate credit to previous studies developing the concepts used by us.

*Choice of CNR Threshold and Confidence Interval: The choice of upper CNR thresholds of -25 dB for WLS200s seem arbitrary, and very conservative. Based on Figure E1, the CNR threshold should be closer to the lower*

*CNR threshold i.e. -30 dB. By arbitrarily setting the upper threshold at -25 dB, a lot of data is categorized as potentially bad, and exaggerates the benefit of the confidence background method presented in the paper. I suggest the authors characterize the noise floor for one of the WLS200s presented in the paper and then assess the benefit of the confidence background method. If a proper noise floor (e.g. -29 dB) had been considered, I think all the gains made during the first iteration shown in Figurer 11a would have been included in the retrieval in the first place. In such a case, the gains from using the confidence interval method would be minimal and could even argued as not worth the effort.*

We agree that $-25\,\mathrm{dB}$ used for the WLS200s is a conservative threshold. Similarly, the $-22\,\mathrm{dB}$ for the StreamLine XR+ and $-5\,\mathrm{dB}$ for the WTX are conservative thresholds for these systems. The purpose of the conservative thresholds is to retain only reliable wind profiles, since the background is used to guide wind profile retrievals at lower CNR levels. Therefore, we placed the threshold at a level for which the presence of noise in the spectrum is very low. For us, this threshold is indicated by the broadening of the radial velocity distribution in the previous Fig. D3, which we now place and discuss more prominently in the manuscript (Fig. 8 in the revised manuscript).

Note that we also present the results of a direct non-iterative retrieval using a -30 dB CNR threshold in Fig. E1, as suggested by you. Fig. E1 shows that the number of retrieved wind profile points is reduced when using a direct CNR thresholded retrieval, compared to our suggested standard module chain using an iterative retrieval. Additionally, individual outliers are present in the edge regions of the retrieval.

For your reference, we also conducted the retrieval for CNR thresholds $-25\,\mathrm{dB}$, $-26\,\mathrm{dB}$, $-27\,\mathrm{dB}$, $-28\,\mathrm{dB}$, $-29\,\mathrm{dB}$, $-30\,\mathrm{dB}$. You can find the results at the end of this document (Fig. 1 - 6). The number of retrieved wind vectors is 4945 for $-25\,\mathrm{dB}$, 5380 for $-29\,\mathrm{dB}$, and 5451 for $-30\,\mathrm{dB}$. In comparison, the proposed standard module chain yields 5556 wind vectors, i.e. more points than the directly thresholded retrieval variants. In addition, the outliers present in the directly thresholded retrievals are avoided.

The determination of a complete 'noise floor' within the retrieval itself is non-trivial and beyond the scope of this study. Noise characteristics depend on the lidar system, laser characteristics (e.g. pulse length/energy, pulse repetition frequency), data acquisition settings (e.g. detection bandwidth, detection algorithm, number of averaged laser pulses, noise level and noise compensation quality), which may vary between different users, systems and measurements. The difficulties, as well as possible solutions, are discussed in [Päschke and Detring, 2024], to whom we now refer in the manuscript.

The conservative thresholds used in the iterative retrieval are only suggested for use in our so-called 'standard module chain'. The standard module chain is not advertised as the most efficient or best possible processing chain, but as an easy to use solution for users with little Doppler lidar processing experience. The processing and retrieval requirements may differ depending on the user and field of application. Our software gives the user the opportunity to set the thresholds individually and/or modify the retrieval chain. We also present and discuss results from a more simple, single iteration, direct CNR threshold, based module chain.

We now state the used thresholds more clearly, label them as conservative and indicate them in the Figure mentioned in the referee comment. Further, an extended discussion on user choices on CNR thresholds and their potential impact is now included. Additionally, the potential gains of the iterative retrieval are put into context.

For your reference, we here include retrieval quality evaluations with different CNR thresholds applied (Fig. 1 - 6.)

*Note that depending upon the definition of the CNR used by manufacturers, and measurement bandwidth, usable CNR threshold for a Doppler lidar will be different. Thus, noise floor for each lidar needs to be characterized independently so that all good data are used in the retrieval.*

*What CNR thresholds were used from the WLS200s at Payerne, WTX at VS and StreamLine XR+ at Neumayer Station? Based on figure D3, the CNR threshold for WLS200s, WTX and StreamLine XR+ should be very different.*

We agree on the importance of varying parameters between systems and apologize for the previously unclear presentation of the used values. Noise characteristics depend on the lidar system, laser characteristics (e.g. pulse length/energy, pulse repetition frequency), data acquisition settings (e.g. detection bandwidth, number of averaged laser pulses, noise level and noise compensation quality), which may vary between different users, systems and measurements. The difficulties, as well as possible solutions, are discussed in [Päschke and Detring, 2024], to which we now refer in the manuscript. For our calculations we used as a conservative threshold for reliable measurements $-25\,\mathrm{dB}$ for WLS200s, $-22\,\mathrm{dB}$ for the StreamLine XR+, and $-5\,\mathrm{dB}$ for the WTX. The second-pass thresholds were set to $-30\,\mathrm{dB}$ for WLS200s, $-30\,\mathrm{dB}$ for the StreamLine XR+, and $-12\,\mathrm{dB}$ for the WTX (Table A2 now). We now state the parameters and values used in our standard module chain in Tab. A2. An advantage and purpose of our software is that different threshold levels can be applied by every user depending on the system and operation characteristics. Since the used module chain and configuration parameters (not limited to the CNR thresholds) are stored in the output metadata, the applied retrieval remains reproducible

and modifiable by others.
We now state the CNR thresholds used for the systems more clearly, placed former Fig. D3 in the main manuscript part and extended the discussion on the selection of the CNR thresholds.

*Some peak detection algorithms used by Doppler Lidars to determine line of sight velocity (LOSV) follow the strongest peak from the high CNR region to lower CNR region. This could potentially result in wrong LOSV determination in presence of wind shear. This could be the case with StreamLine XR+ at Neumayer Station. The confidence background method would potentially double down on determining these erroneous LOSV data points as good data points. So, one needs to be cautious when applying methods such as confidence interval to determine reliable data points in the low CNR regions.*

We obtained the measurements from Neumayer Station from the ACTRIS Cloudnet data portal where no raw spectral data is available, such that we cannot verify this hypothesis. Since we are not the operators of the StreamLine XR+ at Neumayer station, and also don't have experience in operating these systems, we cannot determine the reasons for the observed behavior. Its cause may be software or hardware issues, or interference with other electronic systems (see appendix D). The operators of the system have been informed on our observations and thankfully acknowledged the information. We agree, judging the reliability of measurements at low CNR is challenging. This difficulty is the reason why we conduct an iterative retrieval starting from the above discussed conservative CNR thresholds to construct the confidence background. The proposed algorithm includes an iterative removal of radial velocity measurements which deviate by more than a user determined value from the acceptable background (3 m/s in the standard module chain). Since unreliable measurements at low CNR typically resemble white noise, we do not see an effect of erroneous modification of the conservative confidence background by weak CNR measurements. To validate the appropriate application of the confidence background, we validate the retrieval with independent radiosonde measurements under a variety of conditions, showing high data availability and reliable retrievals also in weak CNR conditions. We have now included an analysis of retrieval errors as a function of CNR, which also does not show an impact of low CNR on retrieval quality (Sect. 5.3 and Fig. 16). The influence of CNR on retrieval quality are discussed in detail in our answer below ('Comparison with radiosondes').

Added impact of CNR in Sect. 5.3 and Fig. 16, see below changes in 'Comparison with radiosondes' answer.

*Uncertainty Analysis: There is no description of the uncertainty in the paper. How is it calculated? How is it impacted by having different types of scans? Depending upon the scans and retrieval grid used, some grids will have more data points than others, how does this impact the uncertainties? Please add how uncertainty is calculated in Appendix C.*

We are not sure to which aspect of uncertainty you are pointing on. Different aspects of uncertainty require consideration in wind profile retrievals:

- Uncertainty due to random errors in the measured radial velocities, related to the CNR discussion above: As discussed, measurement conditions, laser and data acquisition settings impact the amount of random error present in the radial velocities. In the wind profile retrievals, such random errors contribute to the residuals between the wind vector estimated from all measurements and the individual radial velocity measurements. Typically, levels or random radial velocity noise are low (i.e. below 0.5 m/s) and thereby present a smaller source of error compared to turbulent fluctuations in the retrieval volume. Additionally, random errors should present white noise and hence average out overall, if a sufficient number of measurements are used in the retrieval (note that gross outliers are prevented through the iterative removal of radial velocity deviations beyond a user defined threshold, in our case 3 m/s). To reduce the impact of random radial velocity noise, a longer integration time or more measurements within one bin can be considered.

- Numerical uncertainty in the wind profile retrieval process: In Section 2.1, we address the issue of insufficient beam dispersion in the measurements used for wind profile retrieval. To avoid a high impact of error propagation, wind vectors with an insufficient dispersion of the underlying measurements are rejected by the user-specified condition number (CN) threshold (CN > 8 threshold in the standard module chain).

- Lidar representativeness error: The wind at the locations measured by the lidar beam may not correspond to the average wind present within the retrieval volume spanned by the lidar beams during the retrieval time. Unfortunately, this error can only be assessed in LES-based virtual lidar studies where the 3D wind vector is known everywhere in the retrieval volume (see e.g. [Gasch et al., 2020, Gasch et al., 2023, Rahlves et al., 2022, Robey and Lundquist, 2022]).

- Retrieval error due wind field inhomogeneity (e.g. due to turbulence) within the retrieval volume (see e.g. [Gasch et al., 2020, Gasch et al., 2023, Rahlves et al., 2022, Robey and Lundquist, 2022]): Turbulence in the volume probed by the lidar beam causes deviations of the wind retrieval, assumed to be homogeneous in the retrieval process. Fluctuations below the (temporal and spatial) size of the retrieval volume are detectable as deviations of the retrieved, projected wind vector from the measured radial velocities, i.e. residuals similar to the random radial velocity errors. Fluctuations beyond the (temporal and spatial) size of the retrieval volume are *not* detectable as deviations, but cause erroneous mapping of wind components instead. Unfortunately, this error can only be assessed in LES-based virtual lidar studies where the 3D wind vector is known everywhere along the path of the lidar scan (see e.g. [Gasch et al., 2020, Gasch et al., 2023, Rahlves et al., 2022, Robey and Lundquist, 2022]).

- Representativeness error in measurement intercomparison: Differences between the Doppler lidar retrieved wind and reference measurements may arise from differences in the probe volume, besides measurement errors of the reference measurement. See below comment on the radiosonde intercomparison.

Due to the superposition of the above factors influencing retrieval accuracy, an overall uncertainty associated with each individual wind profile is difficult to establish and beyond the scope of this study. For this reason, we chose the validation through intercomparison with radiosondes as a verification method. Especially the lidar representativeness, retrieval, and measurement intercomparison errors, driven by turbulence, are often the largest, but difficult to estimate in real measurements. Estimation of these errors typically requires application of LES-based models where the 3D wind vector at the point of the lidar measurements and within the retrieval volume is known (e.g. [Gasch et al., 2020, Gasch et al., 2023, Rahlves et al., 2022, Robey and Lundquist, 2022]).
The software provides the variance of the residuals as an indicator of the agreement of radial velocities measurements within each bin. This metric enables the user to identify the radial velocity variance level within each retrieval bin, which is often related to random radial velocity error and/or turbulence. However, the metric can only be used for a comparative uncertainty/turbulence assessment since the scan pattern can change within and between retrieval bins.
Having the variance/standard deviation for the single wind components is desirable but not easy to implement. The precision estimation introduced by [Newsom et al., 2017] uses the variance of the residuals for the estimation of the uncertainty in u and v direction (see equations (7) and (1) there). Similarly, the standard module chain provides the variance of the residuals as an indicator (that needs to be used carefully). Using highly frequent measurements for a reliable determination of the uncertainty, e.g. carried out by Steinheuer et al. [Steinheuer et al., 2022], has special requirements for the scan pattern (very short scans with fast scanner rotation). For the proposed standard module chain the intention is to keep the scan pattern as flexible as possible. The modular architecture enables specialized modules, e.g. for evaluations based on special scans. The scope of the present contribution is, however, the introduction of this flexible architecture and a retrieval algorithm (module chain) with a broad applicability, high availability and quality suitable for data assimilation. In case your question addresses another topic or focuses on one of the aspects, please provide additional details. We have now included a more extended discussion of the aspects contributing to uncertainties in the retrieved wind vector in the radiosonde comparison section (Sect. 5.2) of the revised manuscript.

*Comparison with radiosondes do not provide retrieval error for individual wind profiles. Depending upon the atmospheric conditions, some wind profiles will have larger uncertainty than others. How is this calculated and is the calculated uncertainty representative? This is especially important for low CNR regimes. Bulk comparison as shown in the paper usually hides the larger bias and uncertainty likely present in the low CNR regimes. For example, all three stations show a bias in v at higher altitudes (Figure 13, 14 and 15). How does this comparison look if only the low CNR regime data are included? I suggest the authors characterize performance/uncertainty as function of CNR.*
We use radiosondes for validation, since there are no other wind measurements available above the surface. We agree, determining an intercomparison error for each individual wind profile is possible but not meaningful due to the potential retrieval errors listed above ('Uncertainty Analysis'). For this reason we only provide standard statistical error metrics for the bulk intercomparison or individual altitude bins. To ensure representativeness of the intercomparison overall 17 months of data and more than 500 radiosonde profiles at three different sites are used in the intercomparison. As pointed out in Section 5.2 and our above answer, radiosondes are not an ideal reference measurement. As an in-situ measurement they produce reliable measurements for a specific time and specific location. The retrieved Doppler lidar profiles represent a vertical profile of the wind within a time span (in our case 10 minutes) and retrieval volume (depending on the used scan directions and elevations). Hence, the representativeness error cannot be avoided in real-world intercomparisons. However, the added representativeness error means that the underlying deviations between radiosonde and wind lidar are on average smaller than observed in the bulk intercomparison.
To analyze and validate the retrieval error as a function of CNR, we have now included the intercomparison statistics as a function of lidar CNR. Overall, little effect of decreasing lidar CNR on retrieval error is discernible.

Additionally, one has to keep in mind cross-correlations with additional variables. Lower CNR are typically measured above the boundary layer, i.e. at higher altitudes. There the distance between lidar and radiosondes is larger on average, leading to an increased representativeness error.

Providing an uncertainty estimate for each wind profile solely based on Doppler lidar measurement data is non-trivial and beyond the scope of this study, as discussed above ('Uncertainty Analysis'). Previous LES-based studies haven shown that the retrieval errors due to turbulence can overwhelm those due to typical random radial velocity error (see e.g. [Gasch et al., 2020, Gasch et al., 2023, Rahlves et al., 2022, Robey and Lundquist, 2022]), but are difficult to quantify based on the lidar measurements alone.

To enable insight in the single profile intercomparisons, we added plots of all radiosonde-lidar comparisons to the radiosonde comparison dataset (see data availability statement).

We have now included this discussion Sect. 5.3 and Fig. 16 to characterize the performance as a function of CNR. We added plots of all single radiosonde comparisons to the provided dataset.

*Minor Comments:*

*First Guess: It is not appropriate to call the results of the initial VAD fit as first guess. It is a result of the full VAD retrieval, and not a guess. I suggest the authors replace it with initial result or first result.*
We changed it into *initial retrieval/initially retrieved wind profiles* as suggested by reviewer.

*Line 126: "CNR is not a completely reliable indicator". This is a misleading statement. One could argue no variable is a completely reliable indicator. One could always filter out very high CNR data to remove contamination from hard targets or moving objects. I suggest the authors revise this statement to "CNR for each lidar needs to be carefully characterized to determine the appropriate thresholds for reliable data".*
We agree. The revised version is: *"The CNR is system and measurement setup dependent and needs to be carefully characterized to determine the appropriate thresholds for reliable measurement".*
We revised this sentence.

*Line 182: How and why does time coordinate differ between level 0 and level 1 data? It seems level 1 data is just standardizing the data format for different types of lidars?*
This is correct. Level 1 is a standardized data format to homogenize data from different lidar types. The time coordinate between level 0 and level does not differ. An exception is if the timestamp indicated in the level 0 dataset does not represent the centre of the measurement interval and the difference is not negligible. In that case the coordinate of the level 1 dataset could be shifted.
We clarified the meaning of level 0 and level 1 in the revised text.

*Line 188: It is not clear why anyone would want to change the range gate settings for different scans. It seems less practical and adds to complexity to lidar operation, data analysis, and in case of this paper, complexity of description. For simplicity, I suggest the authors use same settings for all scans so that the number of range gates is limited by the pulse repetition frequency (PRF), and all scans use same number of range gates. In that case, Figure 2 will look same as Figure 3 with all the block filled in.*
For DBS scans on the WLS200s system, available from the factory settings, the vertical beam utilizes different range gate lengths than the tilted one. Such a setup enables ordinary retrievals to calculate wind vectors based on range gates, while ensuring availability of a vertical beam measurement within every retrieval volume. Since we operate multiple WLS200s and the DBS scan is a factory setting we have the requirement of dealing with changing range gate numbers within a single scan. Thereby, the software is required to consider different range gate settings even within one scan.
We highlighted in the introduction that users with no influence on the measurement setup (e.g. using the default WLS200s DBS scan), as well as Doppler lidar operators with opportunity for individual settings, are addressed.

*Line 188: I would assume level 0 data is in range gate index, and you add range information in level 1? For example, the StreamLine level 0 data used to include range gate index and not range. Maybe this has changed with newer versions?*
Yes, StreamLine hpl files include range gate indices, but WLS200s include ranges, WLS7 defines vertical altitudes. To standardize and enable homogenized retrieval, we propose the suggested data format.
No changes.

*Line 446: The lower number of retrieved bins in simple -30 dB CNR threshold retrieval due to +/-3 m/s*

*filtering?*
Yes.
We clarified this in the text.

*Figure 7 & 8: It would be better to include Figure 8 as the 4th panel in Figure 7. Figure 8 caption: The data gap is at 21:40 UTC.*
This is a good suggestion. We thank you for the careful reading and corrected this mistake.
We included the figure as a subpanel and corrected the mistake.

*Figure 12: Is the omitted radiosonde profile shown as x?*
Yes.
We added a note in the Figure caption.

**Supplementing figures**

[Figure]

Figure 1: Result of modules (1) to (6) with a CNR threshold of $-25\,\mathrm{dB}$ in module (4). In sum, 4945 wind vectors are retrieved (compared to 5556 vectors when using the proposed standard module chain).

[Figure]

Figure 2: Result of modules (1) to (6) with a CNR threshold of $-26\,\mathrm{dB}$ in module (4). In sum, 5091 wind vectors are retrieved (compared to 5556 vectors when using the proposed standard module chain).

[Figure]

Figure 3: Result of modules (1) to (6) with a CNR threshold of $-27\,\mathrm{dB}$ in module (4). In sum, 5193 wind vectors are retrieved (compared to 5556 vectors when using the proposed standard module chain).

[Figure]

Figure 4: Result of modules (1) to (6) with a CNR threshold of $-28\,\mathrm{dB}$ in module (4). In sum, 5293 wind vectors are retrieved (compared to 5556 vectors when using the proposed standard module chain).

[Figure]

Figure 5: Result of modules (1) to (6) with a CNR threshold of $-29\,\mathrm{dB}$ in module (4). In sum, 5380 wind vectors are retrieved (compared to 5556 vectors when using the proposed standard module chain).

[Figure]

Figure 6: Result of modules (1) to (6) with a CNR threshold of $-30\,\mathrm{dB}$ in module (4). In sum, 5451 wind vectors are retrieved (compared to 5556 vectors when using the proposed standard module chain).

**References**

[Bonin et al., 2017] Bonin, T. A., Choukulkar, A., Brewer, W. A., Sandberg, S. P., Weickmann, A. M., Pichugina, Y. L., Banta, R. M., Oncley, S. P., and Wolfe, D. E. (2017). Evaluation of turbulence measurement techniques from a single Doppler lidar. Atmos. Meas. Tech., 10(8):3021–3039.

[Gasch et al., 2023] Gasch, P., Kasic, J., Maas, O., and Wang, Z. (2023). Advancing airborne Doppler lidar wind profiling in turbulent boundary layer flow – an LES-based optimization of traditional scanning-beam versus novel fixed-beam measurement systems. Atmos. Meas. Tech., 16(22):5495–5523.

[Gasch et al., 2020] Gasch, P., Wieser, A., Lundquist, J. K., and Kalthoff, N. (2020). An LES-based airborne Doppler lidar simulator and its application to wind profiling in inhomogeneous flow conditions. Atmos. Meas. Tech., 13(3):1609–1631.

[Handwerker et al., 2025] Handwerker, J., Barthlott, C., Bauckholt, M., Belleflamme, A., Böhmländer, A., Borg, E., Dick, G., Dietrich, P., Fichtelmann, B., Geppert, G., Goergen, K., Güntner, A., Hammoudeh, S., Hervo, M., Hühn, E., Kaniyodical Sebastian, M., Keller, J., Kohler, M., Knippertz, P., Kunz, M., Landmark, S., Li, Y., Mohannazadeh, M., Möhler, O., Morsy, M., Najafi, H., Nallasamy, N. D., Oertel, A., Rakovec, O., Reich, H., Reich, M., Saathoff, H., Samaniego, L., Schrön, M., Schütze, C., Steinert, T., Vogel, F., Vorogushyn, S., Weber, U., Wieser, A., and Zhang, H. (2025). From initiation of convective storms to their impact — the swabian moses 2023 campaign in southwestern germany. Frontiers in Earth Science, Volume 13 - 2025.

[Newsom et al., 2017] Newsom, R. K., Alan Brewer, W., Wilczak, J. M., Wolfe, D. E., Oncley, S. P., and Lundquist, J. K. (2017). Validating precision estimates in horizontal wind measurements from a Doppler lidar. Atmos. Meas. Tech., 10(3):1229–1240.

[Pichugina et al., 2019] Pichugina, Y. L., Banta, R. M., Bonin, T., Brewer, W. A., Choukulkar, A., McCarty, B. J., Baidar, S., Draxl, C., Fernando, H. J. S., Kenyon, J., Krishnamurthy, R., Marquis, M., Olson, J., Sharp, J., and Stoelinga, M. (2019). Spatial variability of winds and hrrr–ncep model error statistics at three doppler-lidar sites in the wind-energy generation region of the columbia river basin. Journal of Applied Meteorology and Climatology, 58(8):1633 – 1656.

[Päschke and Detring, 2024] Päschke, E. and Detring, C. (2024). Noise filtering options for conically scanning Doppler lidar measurements with low pulse accumulation. Atmos. Meas. Tech., 17(10):3187–3217.

[Rahlves et al., 2022] Rahlves, C., Beyrich, F., and Raasch, S. (2022). Scan strategies for wind profiling with Doppler lidar–an large-eddy simulation (LES)-based evaluation. Atmos. Meas. Tech., 15(9):2839–2856.

[Robey and Lundquist, 2022] Robey, R. and Lundquist, J. K. (2022). Behavior and mechanisms of Doppler wind lidar error in varying stability regimes. Atmos. Meas. Tech., 15(15):4585–4622.

[Steinheuer et al., 2022] Steinheuer, J., Detring, C., Beyrich, F., Löhnert, U., Friederichs, P., and Fiedler, S. (2022). A new scanning scheme and flexible retrieval for mean winds and gusts from doppler lidar measurements. Atmospheric Measurement Techniques, 15(10):3243–3260.

[Tucker et al., 2009] Tucker, S. C., Senff, C. J., Weickmann, A. M., Brewer, W. A., Banta, R. M., Sandberg, S. P., Law, D. C., and Hardesty, R. M. (2009). Doppler lidar estimation of mixing height using turbulence, shear, and aerosol profiles. Journal of Atmospheric and Oceanic Technology, 26(4):673 – 688.